# The Art of Interrogation: Consistency Amplifies Factuality in Spatial Reasoning

**Théo Uscidda** [1] [*]  **Marta Tintore Gazulla** [2]  **Maks Ovsjanikov** [3]  **Federico Tombari** [2]  **Leonidas Guibas** [3]

## Abstract

Current Large Reasoning Models (LRMs) exhibit remarkable general capabilities but significantly underperform in spatial reasoning tasks. Existing approaches treat this gap as a knowledge deficit, relying on supervised fine-tuning (SFT) to ingest labeled spatial data from external vision sources or synthetic engines. In contrast, we argue that for many tasks, spatial reasoning capabilities are already present in pre-trained LRMs but require alignment through logical coherence under geometric 2D and 3D constraints. In this work, we propose a self-supervised reinforcement learning (RL) framework that targets the internal reasoning process without requiring ground-truth annotations. By formalizing the notion of consistency verifiers — reward functions that check for geometric and semantic consistency under transformations — we demonstrate that models can improve their spatial reasoning abilities. We use both image transformations, like flipping, and textual transformations, like swapping the order of objects in the question, and propose a new optimal transport-based RL strategy, OT-GRPO, which is a minimal-matching variant of group relative policy optimization tailored to pairwise verifiers. We show that this label-free consistency training approaches the accuracy of models trained with ground-truth supervision and achieves similar generalization across diverse tasks and data domains.

## 1. Introduction

Spatial reasoning remains a weakness for large reasoning models (LRMs). Recent benchmarks quantify the gap: models underperform humans by 30–40% on tasks such as relative position, depth ordering, and size comparison (Stogiannidis et al., 2025; Yu et al., 2025; Cai et al., 2025b;

[*]Work done during an internship at Google. [1]CREST, ENSAE, Institut Polytechnique de Paris [2]Google Zurich [3]Google DeepMind. Correspondence to: Théo Uscidda <theou@google.com>.

*Proceedings of the $43^{rd}$ International Conference on Machine Learning*, Seoul, South Korea. PMLR 306, 2026. Copyright 2026 by the author(s).

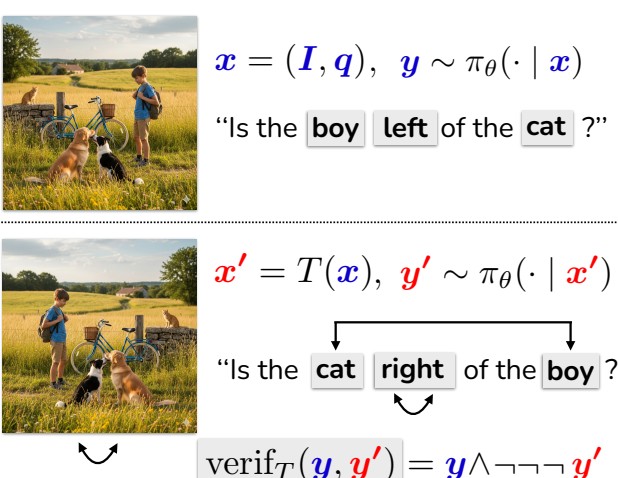

$$\boldsymbol{x} = (\boldsymbol{I}, \boldsymbol{q}), \quad \boldsymbol{y} \sim \pi_\theta(\cdot \mid \boldsymbol{x})$$

"Is the **boy** **left** of the **cat** ?"

$$\boldsymbol{x'} = T(\boldsymbol{x}), \quad \boldsymbol{y'} \sim \pi_\theta(\cdot \mid \boldsymbol{x'})$$

"Is the **cat** **right** of the **boy** ?"

$$\mathrm{verif}_T(\boldsymbol{y}, \boldsymbol{y'}) = \boldsymbol{y} \wedge \neg\neg\neg\, \boldsymbol{y'}$$

*Figure 1.* Example of consistency verifier. Given a prompt asking whether object A is left of object B, we apply transformations (horizontal flip on the image, reformulation on the question) to create an augmented prompt. The consistency verifier checks whether the model's answers satisfy the expected relationship — here, disagreement — without requiring ground-truth labels. For instance, if the model answers `True` on the original and `False` on the augmented prompt, given the transformation, the answers are consistent regardless of the actual spatial arrangement.

Chen et al., 2025a), and on harder compositional tasks only achieve near-random performance (Thrush et al., 2022; Jiang et al., 2025). Failures manifest as high answer variance, sensitivity to question phrasing (Zhang et al., 2024), and systematic violations of geometric laws — for example, a model may answer that object A is to the left of B, and also that B is to the left of A. Reliable spatial reasoning matters for downstream 3D applications in robotics, navigation, and embodied AI, where such inconsistencies are unacceptable.

Existing approaches treat this gap as a factual knowledge deficit. One line of work trains on fully synthetic scenes rendered from 3D simulators (Chen et al., 2024b). Another annotates real images using cascades of vision models — e.g., depth estimators, camera calibrators, open vocabulary detectors — to generate spatial QA pairs, then trains with supervised fine-tuning (Chen et al., 2024a; Cai et al., 2025a; Cheng et al., 2024; Ma et al., 2025). Both approaches aim to compensate for the lack of spatial knowledge by ingesting labeled data from external sources, offering as treatment further training with additional spatial factual supervision.

We take a different view: the necessary spatial capabilities likely exist in pre-trained LRMs, but the model's reasoning process lacks internal coherence and adherence to fundamental geometric principles. Prior work has shown that sampling multiple reasoning paths and selecting the most consistent answer improves accuracy (Wang et al., 2022), or that inconsistency under superficial prompt changes signals unreliable answers (Zhang et al., 2024; Dagan et al., 2025). Spatial reasoning has additional structure that makes consistency appealing. Consider the question "Is the boy to the left of the cat?". Flip the image: left becomes right, so the answer flips. Crop it: the answer stays. Swap the objects in the question: the answer flips again. These transformations have known effects — determined by geometry and language, not scene content — and when models answer consistently under them, they tend to be correct. Crucially, as we know when answers should match or flip, we obtain a self-supervised reward signal without needing ground-truth labels.

We formalize this idea as *consistency verifiers*: reward functions that check whether two model answers satisfy the relationship induced by a transformation. Related ideas of equivariance appear in vision-language alignment (Wang et al., 2023), and in cycle-consistency, which has been used as a reward signal in other domains (Bahng et al., 2025). We optimize consistency verifiers using group-based policy optimization (GRPO) (Guo et al., 2025a). Because our verifier scores *pairs* of completions rather than individual ones, we introduce OT-GRPO, a minimal-consistency matching variant that pairs completions to maximize disagreement. This prevents the model from achieving high reward through lucky alignments and outperforms simpler pairing strategies.

On four spatial reasoning tasks — orientation, depth, size, and relative distance — training with consistency reward alone nearly matches training with ground-truth accuracy reward. This finding suggests that enforcing logical coherence unlocks latent spatial reasoning abilities, without access to labels (Yue et al., 2025; Chen et al., 2025b). We also find that consistency transfers between tasks (e.g., training on orientation improves depth) and between data domains (indoor to outdoor), demonstrating that label-free consistency training generalizes almost as well as ground-truth supervision.

**Contributions.**

- We observe that consistency under geometric and semantic transformations of the prompt is an indicator of correctness in spatial reasoning, and propose to use it as a self-supervised reward signal.

- We formalize *consistency verifiers* — reward functions checking answer consistency under such transformations — and introduce OT-GRPO, a minimal-matching scheme

suited to optimize pairwise rewards with group-based proximal policy optimization methods.

- We show that consistency reward alone approaches accuracy reward on four spatial reasoning tasks (orientation, depth, size, relative distance), and outperforms seven self-supervised baselines including Visual Jigsaw and SSL4RL.

- We find that consistency training transfers across tasks (e.g., depth $\leftrightarrow$ size) and domains (indoor $\leftrightarrow$ outdoor) as effectively as accuracy training, and extends to numeric outputs (counting, absolute distance).

## 2. Related Work

**Spatial Reasoning in VLMs.** A growing body of benchmarks documents the spatial reasoning gap in vision-language models. Mind the Gap (Stogiannidis et al., 2025) evaluates relative position and finds models lag humans by over 30%. SIBench (Yu et al., 2025) tests perspective-taking and metric estimation with similar findings. EASI (Cai et al., 2025b) provides a holistic evaluation across multiple spatial dimensions. They reveal that strong semantic understanding does not transfer to geometric understanding. To close this gap, prior work trains on synthetic scenes from 3D simulators (Chen et al., 2024b) or annotates real images using depth estimators and 3D re-constructors (Chen et al., 2024a; Cai et al., 2025a; Cheng et al., 2024). Both directions require ground-truth supervision. We follow a different path: rather than labeling answers, we exploit the fact that well-selected geometric or textual transformations induce *known* mappings on answers, providing a self-supervised reward signal.

**Consistency as a Proxy for Correctness.** The idea that consistency signals correctness has multiple instantiations. Wang et al. (2022) show that sampling multiple CoT paths and marginalizing over answers (self-consistency) improves reasoning accuracy. Zuo et al. (2025) use the majority answer as a pseudo label for RL training. Zhang et al. (2025) extend this by using cross majority voting across problem reformulations as a self-supervised reward. Zhao et al. (2025) use the entropy of the model as reward signal to favor consistency. Additionally, studies of prompt sensitivity document that models change answers under superficial phrasing changes, suggesting inconsistency as a failure mode (Zhang et al., 2024; Dagan et al., 2025). Concurrent work applies RL to promote logical consistency in VLMs (Anonymous, 2025). In this work we train with consistency reward *only*, without relying on any ground-truth labels.

**Consistency in Computer Vision.** Consistency objectives have a long history in vision. Cycle-consistency enables unpaired image translation (Zhu et al., 2017) and corre-

spondence learning (Zhou et al., 2016; Wang et al., 2019). Recent work uses cycle consistency as a reward for vision-language alignment (Bahng et al., 2025). Our setting differs: we verify a *known* relationship between VQA answers — invariance or equivariance — determined entirely by the transformation design, not learned from data.

**RL for Reasoning.** We build on GRPO (Guo et al., 2025a; DeepSeek-AI, 2025), an RL algorithm that uses verifier-based rewards without a learned critic. The broader role of RL in reasoning remains under debate. Yue et al. (2025) argue that RL with verifiable rewards acts as a filter, improving sampling efficiency without expanding underlying capability. Chen et al. (2025b) suggest RL selects among pre-existing reasoning patterns rather than creating new ones. Our results align with this view: consistency reward steers the model toward answers that satisfy geometric laws, surfacing accuracy already latent in the pretrained model.

## 3. Consistency Verifiers

Standard RL with verifiable rewards (RLVR) requires ground-truth labels to evaluate rewards. We introduce *consistency verifiers*—reward functions that operate without labels by verifying the model's consistency under geometric and textual/semantic transformations between prompts.

**Setup.** Let $\mathcal{D}$ be a dataset of VQA prompts $x = (I, q)$ pairing an image $I$ with a question $q$. Given a prompt $x$, a model $\pi_\theta$ produces a completion $y \sim \pi_\theta(\cdot \mid x)$, which we interpret as the answer. Our goal is to train $\pi_\theta$ to produce correct answers without access to ground-truth labels.

**Prompt Augmentation.** Starting from a prompt $x = (I, q)$, we construct an *augmented* prompt $x' = (I', q')$ by applying a transformation $T$ drawn from a predefined set $\mathcal{T}$. Each transformation $T \in \mathcal{T}$ decomposes as $T = (T_I, T_q)$, where $T_I$ is an image transformation and $T_q$ is a text transformation, yielding $x' = T(x) = (T_I(I), T_q(q))$. Each transformation $T$ induces a *known* mapping $\phi_T$ on the answer space, which constrains how correct answers relate as a pair. Specifically, if $(y^\star, y'^\star)$ are the correct answers to $(x, x')$, then $y'^\star = \phi_T(y^\star)$. The transformation thus determines the relationship within the pair without revealing either answer. Crucially, we know how the pair must relate without knowing the individual values, which enables supervision without ground-truth. Depending on $T$, $\phi_T$ takes one of two forms:

- *Invariance* ($\phi_T = \mathrm{id}$): The transformation preserves the answer. For example, object-preserving crops do not change spatial relationships. A consistent pair must agree.

- *Equivariance* ($\phi_T = \neg$): The transformation changes the answer predictably. For example, a horizontal flip swaps

left and right. Similarly, a text rewrite can swap relational words (e.g., left↔right) or object references (A↔B). A consistent pair must disagree accordingly.

Because $\phi_T$ is fully determined by the choice of $T$, we can check consistency *without* knowing the ground-truth labels.

**Consistency Verifier.** The known mapping $\phi_T$ allows us to check whether completions from related prompts are *mutually consistent*. We define the *consistency verifier* as:

$$\mathrm{verif}_T(y, y') = \begin{cases} 1 & \text{if } y' = \phi_T(y), \\ 0 & \text{otherwise}, \end{cases}$$

where $y \sim \pi_\theta(\cdot \mid x)$ and $y' \sim \pi_\theta(\cdot \mid x')$ are completions from the original and augmented prompts. The verifier returns 1 if and only if the two answers satisfy the expected relationship. Because $\phi_T$ is determined entirely by the transformation design, the verifier can be evaluated without ground-truth labels — it provides a *self-supervised* reward signal based on cross-prompt consistency.

**Illustrative Example.** Consider a VQA where the question is $q$ = `"Is object A left of object B? True/False"`. We construct the augmented prompt $x'$ through the composition of three operations:

1. *Image*: Apply a horizontal flip, which swaps left ↔ right.

2. *Text*: Swap the object references (A ↔ B).

3. *Text*: Replace the relation (left → right).

Each operation negates the answer. Since we apply three negations, the net effect is also a negation: $\phi_T(y) = \neg y$. The key insight is that we can verify consistency *without knowing the ground truth*. If the model answers `True` on $x$ and `False` on $x'$, or vice-versa, the answers are consistent — regardless of whether object A is actually left of object B. In other words, the verifier simply checks disagreement: $\mathrm{verif}_T(y, y') = \mathbf{1}\{y \neq y'\}$ (see Figure 1).

## 4. Learning with Consistency Verifiers

Having defined the consistency verifier, we now turn to optimization. We build on GRPO (Guo et al., 2025a; DeepSeek-AI, 2025). Standard GRPO assumes a scalar reward for each completion, but our consistency verifier scores *pairs* — one completion from $x$ and one from its augmentation $x'$. The main challenge is therefore to convert pairwise consistency scores into per-completion rewards. We recall standard GRPO, then introduce our adaptation.

### 4.1. Background on GRPO

In the standard supervised RLVR setting, we have access to ground-truth answers $y^\star$ and seek to maximize expected

accuracy:

$$\max_{\theta} \ \mathbb{E}_{x\sim\mathcal{D}} \ \mathbb{E}_{y\sim\pi_\theta(\cdot|x)}\big[\mathrm{verif}(y, y^\star)\big], \qquad (1)$$

where $\mathrm{verif}(y, y^\star) = \mathbf{1}\{y = y^\star\}$ is the accuracy verifier. GRPO optimizes this objective by sampling $K$ completions from $\pi_{\theta_{\mathrm{old}}}$ (a snapshot of the current policy):

$$y_{1:K} := (y_1, \ldots, y_K) \sim \pi_{\theta_{\mathrm{old}}}(\cdot \mid x).$$

Each completion receives a reward $r_i = \mathrm{verif}(y_i, y^\star)$, which GRPO normalizes within the group to form advantages:

$$A_i := \frac{r_i - \mathrm{mean}(r_{1:K})}{\mathrm{std}(r_{1:K})}. \qquad (2)$$

Each completion $y_i$ is a token sequence $(y_{i,1}, \ldots, y_{i,T_i})$. GRPO optimizes a PPO-style clipped surrogate at the token level. The importance ratio and clipping operator are:

$$\rho_{i,t}(\theta) := \frac{\pi_\theta(y_{i,t} \mid x, y_{i,<t})}{\pi_{\theta_{\mathrm{old}}}(y_{i,t} \mid x, y_{i,<t})}, \qquad (3)$$

$$\mathrm{clip}_\varepsilon(u) := \min\big(\max(u, 1-\varepsilon), 1+\varepsilon\big). \qquad (4)$$

The token-level surrogate and the full GRPO objective are:

$$\ell_{i,t}(\theta; A_i) := A_i \cdot \min\big(\rho_{i,t}(\theta), \ \mathrm{clip}_\varepsilon(\rho_{i,t}(\theta))\big), \quad (5)$$

$$J_{\mathrm{GRPO}}(\theta) := \frac{1}{K}\sum_{i=1}^{K}\frac{1}{T_i}\sum_{t=1}^{T_i}\ell_{i,t}(\theta; A_i). \qquad (6)$$

## 4.2. From Ground-Truth to Pairwise Consistency

With ground-truth labels, each completion $y_i$ has a canonical "partner" — the label $y^\star$ — against which it is evaluated. In our setting, we do not have access to $y^\star$. Instead, we have paired prompts $(x, x')$ where $x' = T(x)$ for some transformation $T$, and the consistency verifier $\mathrm{verif}_T(y, y')$ that checks whether completions $y \sim \pi_\theta(\cdot \mid x)$, $y' \sim \pi_\theta(\cdot \mid x)$ satisfy the expected relationship induced by the transformation. Our training objective becomes:

$$\max_{\theta} \ \mathbb{E}_{\substack{x\sim\mathcal{D}, T\sim\mathcal{T} \\ x'=T(x)}} \ \mathbb{E}_{\substack{y\sim\pi_\theta(\cdot|x) \\ y'\sim\pi_\theta(\cdot|x')}} \big[\mathrm{verif}_T(y, y')\big]. \quad (7)$$

To use GRPO, we sample $K$ completions from each prompt, yielding a $K \times K$ matrix of verifier scores $\mathrm{verif}_T(y_i, y'_j)$, $1 \le i, j \le K$. The challenge is to reduce this matrix to per-completion rewards $r_i$ for the GRPO update.

**Natural pairing strategies.** Given the verifier matrix, two natural strategies aggregate it into per-completion rewards:

- *Random pairing* pairs completions arbitrarily, e.g., by generation order: $r_i^{\mathrm{rand}} = \mathrm{verif}_T(y_i, y'_i)$.

- *One-to-all* averages over the other group: $r_i^{\mathrm{all}} = \frac{1}{K}\sum_{j=1}^{K}\mathrm{verif}_T(y_i, y'_j)$.

Both yield the same expected reward ($\mathbb{E}[r_i^{\mathrm{rand}}] = \mathbb{E}[r_i^{\mathrm{all}}]$), but one-to-all reduces variance through averaging.

---

**Algorithm 1** One OT-GRPO Iteration (batch size 1).

---
1: **Require:** original prompt $x \sim \mathcal{D}$.
2: **Transform:** sample $T \sim \mathcal{T}$, and set $x' \leftarrow T(x)$.
3: **Generate:** $y_{1:K} \sim \pi_{\theta_{\mathrm{old}}}(\cdot \mid x)$ and $y'_{1:K} \sim \pi_{\theta_{\mathrm{old}}}(\cdot \mid x')$.
4: **Score** (each pair): $V_{ij} \leftarrow \mathrm{verif}_T(y_i, y'_j)$.
5: **Minimal Match:** $\sigma^\star \leftarrow \arg\min_{\sigma\in\mathcal{S}_K} \sum_i V_{i,\sigma(i)}$.
6: **Rewards:** $r_i \leftarrow V_{i,\sigma^\star(i)}$, and $r'_{\sigma^\star(i)} \leftarrow r_i$.
7: **GRPO Updates** on $(y_{1:K}, r_{1:K})$ and $(y'_{1:K}, r'_{1:K})$.

---

## 4.3. Adversarial Consistency Pairing

Let $\pi_\theta^x := \pi_\theta(\cdot \mid x)$ and $\pi_\theta^{x'} := \pi_\theta(\cdot \mid x')$ denote the output distributions on the original and augmented prompts. The two strategies above correspond to sampling $y$ and $y'$ independently, namely $(y, y') \sim \pi_\theta^x \otimes \pi_\theta^{x'}$. More generally, we can sample $(y, y')$ from any *coupling* $\gamma \in \Gamma(\pi_\theta^x, \pi_\theta^{x'})$, i.e., any joint distribution with $\pi_\theta^x$ and $\pi_\theta^{x'}$ as marginals. This yields a generalized objective:

$$\max_{\theta} \ \mathbb{E}_{\substack{x\sim\mathcal{D}, T\sim\mathcal{T} \\ x'=T(x)}} \ \mathbb{E}_{(y,y')\sim\gamma}\big[\mathrm{verif}_T(y, y')\big]. \qquad (8)$$

**Adversarial pairing.** Now, the question is: which coupling should we choose? We propose to be adversarial: select the coupling that *minimizes* the expected consistency. This corresponds to finding an optimal transport (OT) coupling (Peyré & Cuturi, 2019), transforming the objective into a max-min problem:

$$\max_{\theta} \ \mathbb{E}_{\substack{x\sim\mathcal{D}, T\sim\mathcal{T} \\ x'=T(x)}} \ \min_{\gamma\in\Gamma(\pi_\theta^x, \pi_\theta^{x'})} \ \mathbb{E}_{(y,y')\sim\gamma}\big[\mathrm{verif}_T(y, y')\big].$$
$$(9)$$

In practice, we approximate the OT coupling from the $K$ completions sampled from each distribution $\pi_\theta^x$ and $\pi_\theta^{x'}$. This yields a third strategy to aggregate the verifier matrix into per-completion rewards. By the Birkhoff–von Neumann theorem (Birkhoff, 1946), the empirical OT coupling is realized by a permutation—the one minimizing total consistency. We then define the reward as:

$$\sigma^\star \in \arg\min_{\sigma\in\mathcal{S}_K} \sum_{i=1}^{K} \mathrm{verif}_T(y_i, y'_{\sigma(i)}),$$
$$r_i^{\mathrm{OT}} = \mathrm{verif}_T(y_i, y'_{\sigma^\star(i)}).$$

This is a linear assignment problem, solvable in $O(K^3)$ time—negligible for $K \le 16$, the range we use in practice. In Section A.2, we show that (9) is equivalent to minimizing the Wasserstein distance (Santambrogio, 2015) between $\pi_\theta^x$ and $\pi_\theta^{x'}$, with cost $c_T = -\mathrm{verif}_T$, namely the *inconsistency*.

**OT-GRPO loss.** We derive a GRPO surrogate for the max-min objective (9). Each matched pair $(y_i, y'_{\sigma^\star(i)})$ receives the same reward $r_i^{\mathrm{OT}} = \mathrm{verif}_T(y_i, y'_{\sigma^\star(i)})$, normalized into a

shared advantage $A_i$. We apply the standard GRPO update to both completions:

$$J_{\text{OT-GRPO}}(\theta) = \underbrace{\frac{1}{K}\sum_{i=1}^{K}\frac{1}{T_i}\sum_{t=1}^{T_i}\ell_{i,t}(\theta; A_i)}_{\text{completions to orginal prompt } y_i}$$
$$\underbrace{\frac{1}{K}\sum_{i=1}^{K}\frac{1}{T'_{\sigma^\star(i)}}\sum_{t=1}^{T'_{\sigma^\star(i)}}\ell'_{\sigma^\star(i),t}(\theta; A_i)}_{\text{completions to augmented prompt } y'_{\sigma^\star(i)}}.$$

(10)

We treat $\sigma^\star$ as constant when computing gradients. This is justified by Danskin (1966)'s theorem, since $\sigma^\star$ is the minimizer of the inner problem. We stress that both original and augmented prompts contribute to the training loss — answers to augmented prompts do not merely provide pseudo-labels. Alg. 1 summarizes one OT-GRPO iteration.

**Why Minimal Consistency?**  Minimal consistency finds the *most challenging* pairing. If a model produces inconsistent completions, minimal consistency will find and expose those inconsistencies, whereas random pairing might miss them by luck. Conversely, if a model is truly consistent, minimal consistency cannot expose disagreement because none exists. This intuition can be made precise. Under a random baseline where the model guesses uniformly at random, random pairing and one-to-all both yield expected reward $\mathbb{E}[r_i^{\text{rand}}] = \mathbb{E}[r_i^{\text{all}}] = 1/2$. Minimal consistency, however, yields $\mathbb{E}[r_i^{\text{OT}}] \approx 1/\sqrt{\pi K}$, which vanishes as $K$ grows (see Section A for derivation). In our label-free setting, this makes minimal consistency robust to reward hacking: high rewards require genuine consistency everywhere.

### 4.4. Reasoning format

We promote CoT reasoning (Wei et al., 2023) by prompting the model with a system instruction adapted from DeepSeek-R1 (DeepSeek-AI, 2025). Completions are expected to follow a structured format: reasoning enclosed in `<think>...</think>` tags, followed by an answer in `<answer>...</answer>` tags. In addition to the consistency reward, we apply a format reward that checks whether the completion adheres to this structure. We use a weight of 1.0 for both consistency and format rewards.

## 5. Experiments

### 5.1. Experimental Setup

**Tasks.**  We evaluate on four binary VQA tasks covering complementary spatial dimensions: *orientation* (left/right), *depth* (closer/further to camera), *size* (3D volume/metric comparison), and *relative distance* (which object is closer

*Table 1.* Total number of examples per task and domain.

| Task | KITTI | SUN RGB-D |
|------|-------|-----------|
| Orientation | 2,406 | 7,738 |
| Depth | 4,288 | 6,222 |
| Size | 3,996 | 4,845 |
| Rel. Distance | 3,539 | 7,015 |

to an anchor). Each task highlights objects in the image and asks a True/False question. Ground truth comes from 3D bounding box annotations; see Section D for details.

**Data.**  We use images from KITTI (outdoor driving) and SUN RGB-D (indoor scenes) with Omni3D 3D annotations (Brazil et al., 2023). To ensure unambiguous ground truth, we filter individual objects by visibility, size, and camera-relative position, and pairs by bounding-box overlap and a minimum metric gap along the task-relevant dimension; remaining pairs are scored by separation and sampled to balance diversity with unambiguity. Each task uses five paraphrased question templates sampled at training time. Set sizes are provided in Table 1; see Sections D and D.5 for full details and example prompts. All models train on SUN RGB-D; KITTI is held out to test domain generalization.

**Transformations.**  We construct augmented prompts by applying image and text transformations, each sampled independently with probability 0.5. Image transforms include horizontal flip, object-preserving crop, and color jitter. Text transforms include swapping object order and swapping the queried relation (e.g., left $\leftrightarrow$ right). Some transforms are *invariant* (answer unchanged); others are *equivariant* (answer flips). The consistency verifier uses the known effect of each transform to determine whether answers should match.

**Models.**  We fine-tune Qwen2.5-VL 3B and 7B with LoRA (Hu et al., 2021) with $r = 32$ and $\alpha = 64$ applied to attention and MLP layers. The vision encoder is frozen.

**Training.**  Each task/source uses an 80/20 split with a fixed seed shared across experiments. Because dataset sizes vary (Table 1), we fix training steps rather than epochs: $T = 500$ steps with $\eta = 10^{-6}$, group size $K = 8$, $\tau = 1.0$, and no KL penalty ($\beta = 0$). Each step samples 4 prompt pairs per device across 8 H100 GPUs (256K completions total). For minimal pairing we use the network-simplex solver from POT (Flamary et al., 2021).

**Evaluation.**  We evaluate on the held-out 20% test split of every (task, source) combination and use two training regimes. In the *per-task* regime (Figures 2 to 4, 7 and 8), each model is trained on a single task's SUN RGB-D training set and evaluated on the test splits of all (task, source)

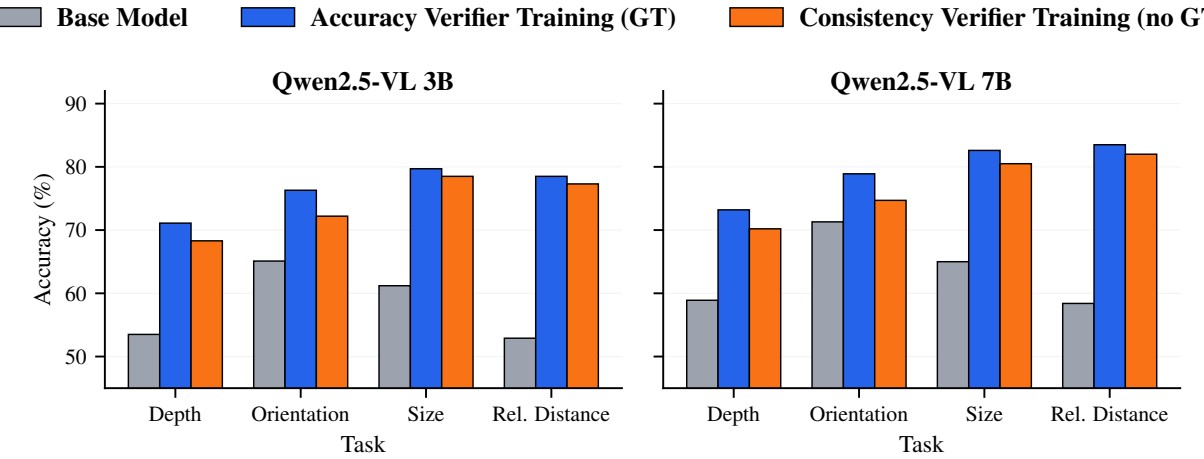

*Figure 2.* Same-task evaluation accuracy on SUN RGB-D (indoor) data for two model sizes (3B and 7B) and four tasks (Depth, Orientation, Size and Relative Distance). We separately train models using either an accuracy verifier (requires ground-truth labels) or a consistency verifier (no labels needed), then evaluate on held-out test samples. Despite never seeing any ground-truth labels during training, consistency-trained models nearly match accuracy-trained ones, with average gaps of only 2.3pp (3B) and 2.7pp (7B).

pairs, yielding the same-task, cross-task, and cross-domain measurements reported below. In the *joint* regime (Figures 5 and 6), our model is trained on the union of the four boolean SUN RGB-D training sets; we then report the average test accuracy across the four SUN RGB-D tasks, with KITTI held out. For every self-supervised baseline appearing in Figure 5 (Visual Jigsaw and SSL4RL variants), we use the publicly released Hugging Face checkpoint without further fine-tuning on our data.

**Accuracy vs. Consistency Training.** We compare two reward signals: *accuracy training* rewards correct answers using ground-truth labels (1/0), while *consistency training* hides labels entirely and rewards predictions that satisfy the expected consistency relationship under transformations. For fairness, both methods share the same data, the same online-sampled image and text augmentations, and the same held-out test set.

### 5.2. In-Depth Comparison: Accuracy vs. Consistency

**Consistency approaches ground-truth accuracy.** Figure 2 compares baseline, accuracy training, and consistency training on SUN RGB-D. Consistency closely tracks accuracy: 76.8% vs. 79.6% for 7B (gap 2.8pp) and 74.1% vs. 76.4% for 3B (gap 2.3pp), both well above baseline (63.4% / 58.2%). The gap is stable across tasks (2.1–3.4pp for 7B), showing that the self-supervised consistency signal recovers most of the gain achievable from labels.

**Consistency transfers across tasks.** Figure 3 reports cross-task transfer (7B, SUN RGB-D), with cells showing improvement over baseline by training/eval task. All 12

off-diagonal entries are positive for both methods (worst: orientation→size, +5.8pp), giving average gains of +10.1pp (accuracy) vs. +9.2pp (consistency); same-task (diagonal) gains are +14.3pp vs. +13.5pp—a sub-1pp gap throughout. Transfer is asymmetric: depth yields the largest average cross-task gain (+12.3 / +11.1pp) while orientation gives the least (+7.8 / +7.2pp). This is consistent with depth being the only task whose answer depends on a continuous 3D axis that also underlies size and inter-object distance comparisons, while orientation is essentially a 2D judgment whose representations transfer least to the other three tasks.

**Consistency transfers across tasks and domains.** Figure 4 shows transfer from SUN RGB-D (indoor) to KITTI (outdoor); diagonals capture same-task domain shift, off-diagonals combine task and domain shift. Same-task gains over the KITTI baseline are +15.6pp (accuracy) vs. +14.7pp (consistency); cross-task gains are +10.9 vs. +10.5pp— again a sub-1pp gap. Despite a higher KITTI baseline (68.2% vs. 63.4% for 7B), same-task cross-domain transfer (+15pp) exceeds same-task within-domain gains on SUN RGB-D (+13.5pp), suggesting spatial reasoning learned on cluttered indoor scenes generalizes well to sparser outdoor environments and that task alignment matters more than domain similarity.

### 5.3. Comparison to Self-Supervised Baselines

**Consistency outperforms self-supervised baselines.** Across both families of self-supervised baselines, label-free consistency wins outright (Figure 5). To benchmark against published methods that share a task pool, we train jointly on the four boolean tasks (Qwen2.5-VL-7B, 2,000

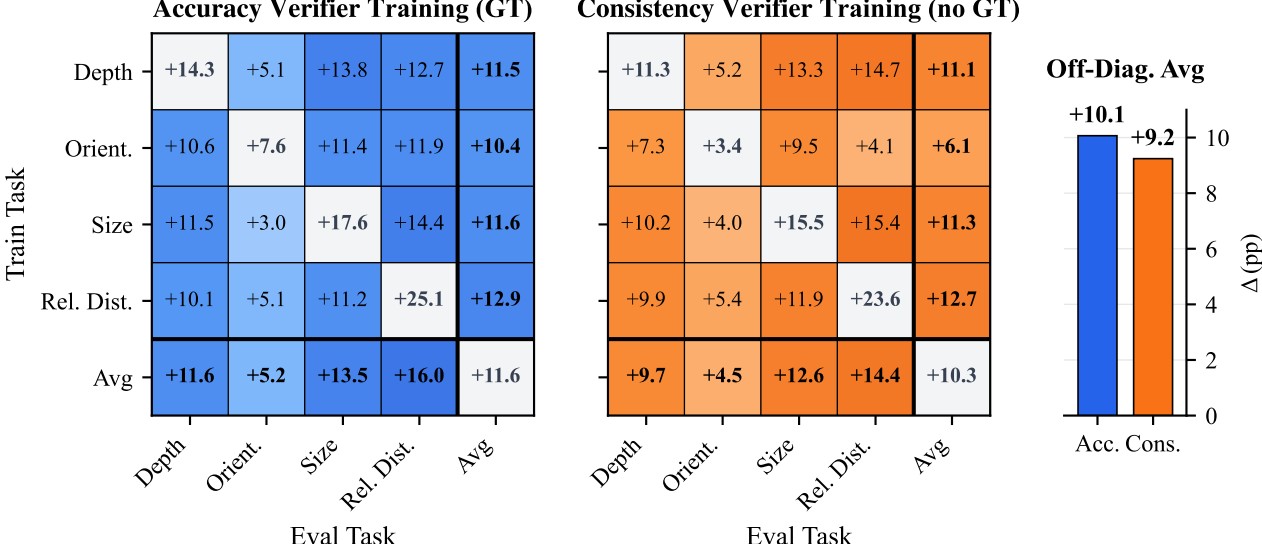

*Figure 3.* Cross-task transfer on SUN RGB-D (7B model). Each cell shows the improvement over the pre-training baseline (in percentage points) when training on the row task and evaluating on the column task. Off-diagonal cells (colored) show cross-task transfer; diagonal cells (gray) show same-task performance for reference. The rightmost panel shows the average off-diagonal improvement: consistency training nearly matches accuracy training with a gap of only 0.8pp.

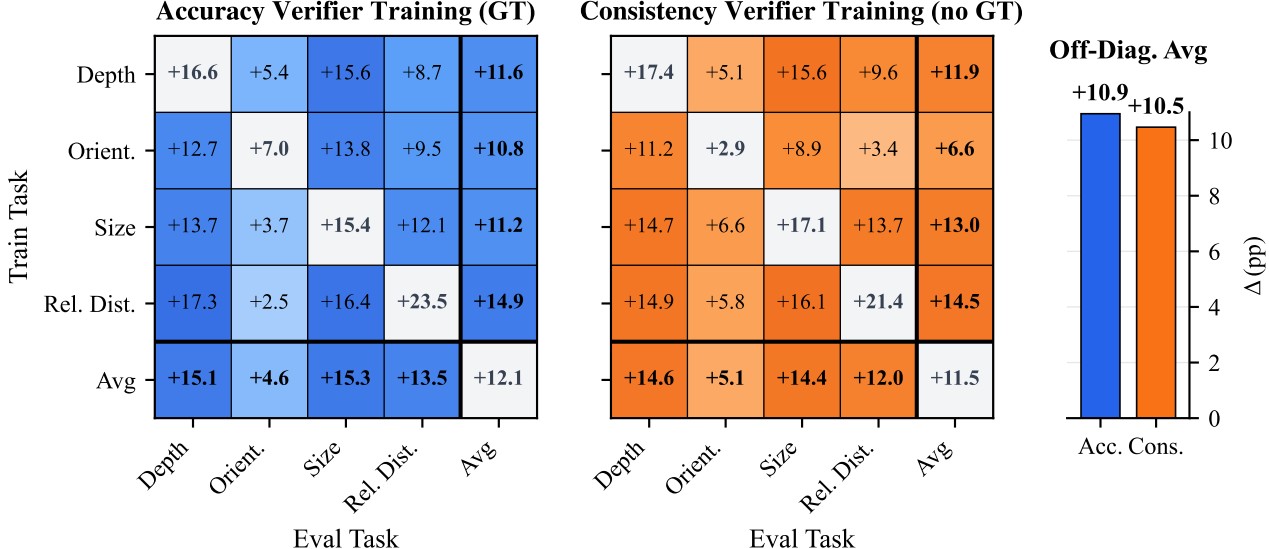

*Figure 4.* Cross-domain transfer from SUN RGB-D to KITTI (7B model). Models are trained on indoor scenes (SUN RGB-D) and evaluated on outdoor driving scenes (KITTI). Diagonal cells (gray) show same-task cross-domain transfer; off-diagonal cells (colored) combine both task and domain shifts. Despite the visual gap between indoor and outdoor environments, both training methods generalize well. Consistency training nearly matches accuracy training, with off-diagonal gaps of only 0.5pp.

steps on ∼40K shuffled SUN RGB-D examples, all other hyperparameters as above) and compare against the Visual Jigsaw and SSL4RL checkpoints released by the authors on Hugging Face. At 83.8%, our reward beats the strongest Visual Jigsaw variant (Wu et al., 2025) (Image, Video, 3D) by +3.6pp and four SSL4RL variants (Guo et al., 2025b)

(Rotation, Position, Contrastive, Jigsaw) by +13–24pp; accuracy training (with labels) tops the table at 86.5%, leaving only a 2.7pp gap to label-free consistency. Rotation/jigsaw-style pretext tasks transfer the worst—several land below the 60.6% pre-training baseline.

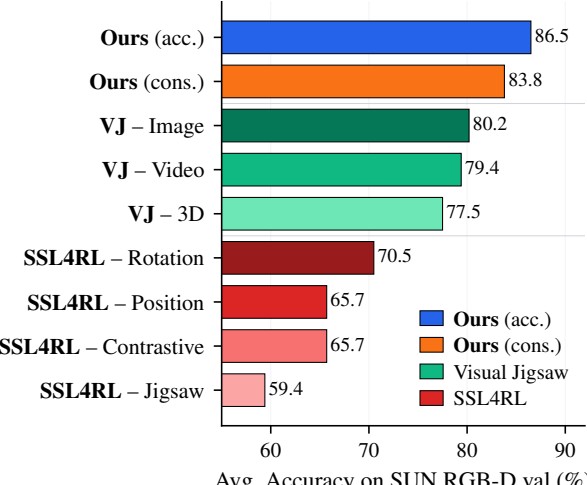

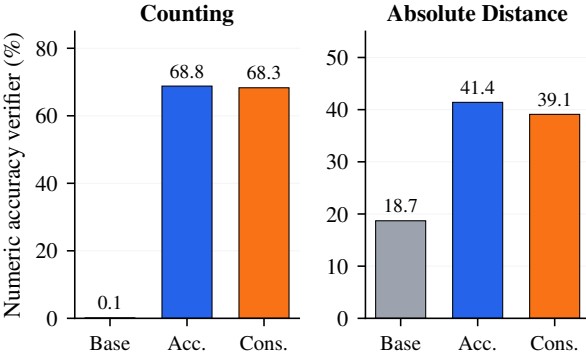

*Figure 7.* Extension to numeric tasks. Numeric accuracy verifier $\text{verif}(y, y^\star) = \max(0, 1 - |y - y^\star|/y^\star)$, reported as a percentage, on counting (integer object counts) and absolute distance estimation (meters). Consistency closely tracks accuracy on both tasks, trailing by 0.5pp on counting and 2.3pp on absolute distance.

*Figure 5.* Comparison against self-supervised baselines on SUN RGB-D (val), averaged over the four boolean tasks. Our method (acc. and cons.) uses Qwen2.5-VL-7B trained jointly on the four boolean tasks (2,000 steps); Visual Jigsaw and SSL4RL variants are evaluated from their publicly released Hugging Face checkpoints, without further fine-tuning on our data. Our consistency reward (no labels) outperforms every self-supervised baseline and trails accuracy training (with labels) by only 2.7pp.

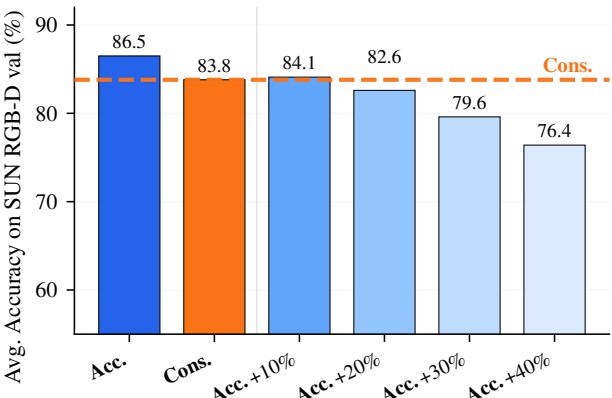

*Figure 6.* Robustness to label corruption. "Acc. $+N\%$" flips $N\%$ of training labels uniformly at random in accuracy training; consistency training uses no labels at all (orange dashed line). Accuracy still edges out consistency at 10% corruption, but the label-free signal overtakes it from 20% onward.

### 5.4. Robustness to Label Corruption

**Consistency overtakes accuracy from 20% noise.** Reusing the all-tasks protocol above, we corrupt accuracy training by flipping each ground-truth label independently with probability $p \in \{0.1, 0.2, 0.3, 0.4\}$ before computing the reward; consistency, reading no labels, is unaffected. Clean accuracy reaches 86.5% and consistency 83.8%. At 10% corruption accuracy drops to 84.1%, essentially matching the label-free signal (within 0.3pp); from 20% onward

consistency overtakes accuracy outright, with gaps of 1.2pp (Acc. 82.6%), 4.2pp (79.6%), and 7.4pp (76.4%) at 20%, 30%, and 40% noise, respectively (Figure 6). Our uniform per-example flips are a conservative noise model: real annotation pipelines tend to produce errors that are *correlated* and concentrated on the harder examples, so the crossover where consistency overtakes accuracy would likely arrive below the 20% reported here. This matters in practice: spatial-reasoning annotation pipelines chain depth estimators, calibrators, detectors, and frontier LLMs (Chen et al., 2024a; Cheng et al., 2024), so even small per-stage error rates compound into corruption levels where consistency overtakes supervision.

### 5.5. Exploration on Numeric Tasks

**Consistency closely matches accuracy on numeric outputs.** We extend the framework to two numeric tasks on SUN RGB-D: *counting* (integer in $\{2, \ldots, 5\}$, e.g., `How many chairs are in the image?`) and *absolute distance estimation* (continuous meters, e.g., `What is the distance between object 1 and object 2 in meters?`; examples in Section D.7). We reuse the same image and text augmentations introduced for the boolean tasks but drop relation swap, since neither a count nor a metric distance admits an equivariant relation to negate. All retained transformations are therefore *invariant*: jitter, mirroring, cropping around the queried objects, and paraphrasing the question leave the underlying count or 3D distance unchanged. In this regime the consistency verifier collapses to a single invariance check—the model should produce the same answer on both slots of a pair.

Because the prediction is now a continuous quantity rather than a True/False label, both the accuracy and the consistency verifiers from Section 3 have to be adapted: exact

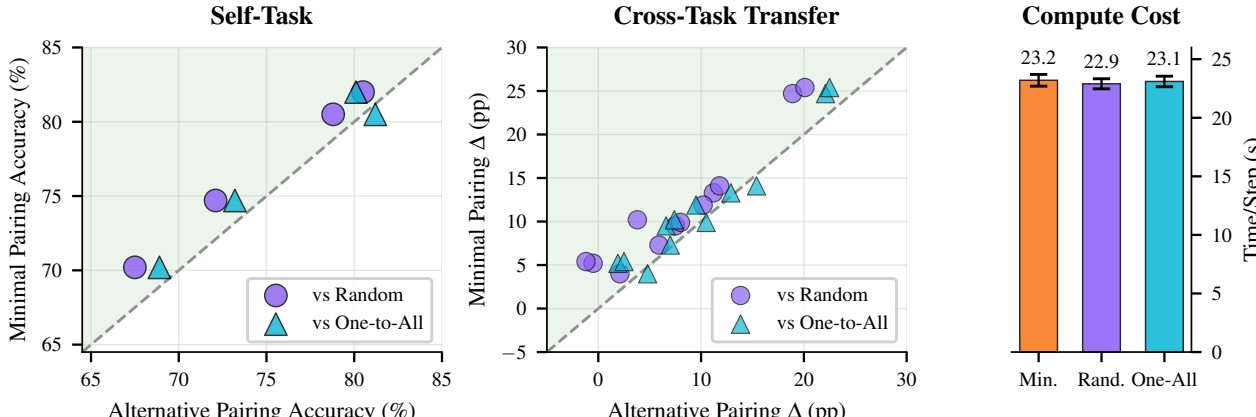

*Figure 8.* Ablation on pairing strategy for consistency training (7B model, SUN RGB-D). Each point compares minimal pairing (y-axis) against an alternative strategy (x-axis): random (purple circles) or one-to-all (cyan triangles). Points above the diagonal indicate minimal pairing outperforms the alternative. **Left:** Self-task accuracy—each point is one of the four tasks, showing that minimal pairing achieves higher accuracy. **Right:** Cross-task transfer improvement over baseline ($\Delta$ in percentage points)—each point is one of the 12 off-diagonal train/eval task pairs, showing that minimal pairing yields larger transfer gains. The green region highlights where minimal pairing wins.

equality is too strict, so we replace it with a *rescaled mean absolute error*. Each verifier returns 1 when its two arguments coincide, decreases linearly in their absolute difference, and clips to 0 once that difference exceeds the reference value. The accuracy verifier rescales the error by the ground truth $y^\star$, while the (symmetric) consistency verifier rescales by the larger of the two predictions, keeping both scores in $[0, 1]$:

$$\text{verif}(y, y^\star) = \max\big(0,\ 1 - |y - y^\star|/y^\star\big),$$
$$\text{verif}_T(y, y') = \max\big(0,\ 1 - |y - y'|/\max(y, y')\big).$$

We train Qwen2.5-VL-7B per task for 500 steps with all other hyperparameters identical to the boolean setting. Consistency closely tracks accuracy on both tasks (Figure 7): on counting, accuracy reaches 68.8% and consistency 68.3%—a 0.5pp gap from a base near chance (0.1%); on absolute distance, accuracy hits 41.4% versus 39.1% for consistency, a 2.3pp gap. In both cases the label-free signal trails the labeled one by less than the boolean-task gap (2.7pp), suggesting the consistency reward remains competitive as we extend beyond binary answer spaces.

### 5.6. Ablation: Pairing Strategy

**Minimal pairing outperforms alternatives.** Figure 8 compares minimal OT pairing to random and one-to-all pairing (7B, SUN RGB-D). Minimal pairing wins on both axes: self-task accuracy of 76.8% vs. 74.7% (random) and 75.8% (one-to-all), and cross-task transfer of +11.7pp vs. +8.2pp and +10.3pp—an advantage that holds across all 4 self-task and 12 cross-task settings. The cost is negligible: 23.2s/step vs. 22.9s and 23.1s, under 1.5% overhead. The gain stems from harder negative pairs: matching each com-

pletion with its most challenging counterpart makes random agreement unlikely (see Section A).

## 6. Conclusion

We introduced consistency verifiers for post-training VLMs on spatial reasoning without ground-truth labels. By exploiting known relationships between answers under geometric and semantic transformations, our approach provides a self-supervised reward signal that approaches the performance of ground-truth supervision — establishing the value of RL post-training for key invariances, equivariances and adherence to general spatial reasoning principles. Experiments demonstrate consistent benefits and generalization across four tasks (orientation, depth, size, relative distance) and two data domains (indoor, outdoor).

**Limitations.** Our consistency rewards compare two predictions at a time. Extending to richer relational structures—chains of three or more transformations, or compositional reasoning steps—could exploit additional geometric structure. Beyond spatial reasoning, applying consistency verifiers to other modalities is a natural next step.

## Impact Statement

This work presents a self-supervised method for improving spatial reasoning in VLMs by exploiting consistency under geometric transformations. The approach reduces reliance on labeled data, which may lower annotation costs and broaden access to model improvement. We do not anticipate specific negative societal consequences beyond those generally associated with advances in machine learning.

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

# A. Further Analysis of the Minimal Consistency Pairing

## A.1. Consistency Pairing Under a Random Baseline

This appendix extends the analysis of pairing strategies introduced in Section 4. We investigate how different strategies behave under an *uninformative* model—one that guesses uniformly at random, ignoring the transformation relationship between prompts. By deriving the expected per-completion reward $\mathbb{E}[r_i]$ under this random baseline, we compare how permissive each strategy is: high reward under random guessing indicates susceptibility to reward hacking, while low reward indicates a stronger learning signal.

**Setup and Notation.** Let $x$ be an original prompt and $x' = T(x)$ an augmented version obtained by applying a transformation $T$. We sample $K$ binary completions from each prompt:

$$y_1, \ldots, y_K \sim \pi_\theta(\cdot \mid x), \qquad y'_1, \ldots, y'_K \sim \pi_\theta(\cdot \mid x'),$$

where each completion $y_i, y'_j \in \{\text{True}, \text{False}\}$.

As described in Section 3, each transformation $T$ induces a known mapping $\phi_T$ on the answer space: if $y^\star$ is the correct answer to $x$, then $\phi_T(y^\star)$ is the correct answer to $x'$. For binary tasks, $\phi_T$ is either the identity (invariant transformations, where answers should match) or negation (equivariant transformations, where answers should differ). The consistency verifier rewards completions that satisfy the expected relationship:

$$\text{verif}_T(y, y') = \mathbf{1}\{y' = \phi_T(y)\} = \begin{cases} \mathbf{1}\{y = y'\} & \text{if } \phi_T = \text{id (invariant)}, \\ \mathbf{1}\{y \neq y'\} & \text{if } \phi_T = \neg \text{ (equivariant)}. \end{cases}$$

For the analysis below, we consider the equivariant case $\text{verif}_T(y, y') = \mathbf{1}\{y \neq y'\}$ without loss of generality; the invariant case is symmetric since $\mathbb{P}(y = y') = \mathbb{P}(y \neq y') = 1/2$ under independent unbiased bits.

**Pairing Strategies.** Given $K$ completions from each prompt, we have a $K \times K$ matrix of verifier scores $V_{ij} = \text{verif}_T(y_i, y'_j)$. Different strategies aggregate this matrix into per-completion rewards $r_i$.

**Random Pairing.** Pair completions by generation order—the $i$-th completion from $x$ with the $i$-th completion from $x'$. The per-completion reward is:

$$r_i^{\text{rand}} = \text{verif}_T(y_i, y'_i).$$

This strategy is simple but arbitrary: the reward depends on the sampling order rather than the content of the completions.

**One-to-All.** Compare each completion from $x$ to every completion from $x'$ and average:

$$r_i^{\text{all}} = \frac{1}{K} \sum_{j=1}^{K} \text{verif}_T(y_i, y'_j).$$

Averaging over all pairs reduces variance but can dilute the signal: a few strong inconsistencies may be hidden when most pairs happen to be consistent.

**Minimal Consistency (OT-GRPO).** Find the permutation $\sigma^\star \in \mathcal{S}_K$ (the symmetric group on $K$ elements) that *minimizes* the total consistency score:

$$\sigma^\star \in \arg\min_{\sigma \in \mathcal{S}_K} \sum_{i=1}^{K} \text{verif}_T(y_i, y'_{\sigma(i)}), \quad r_i^{\text{OT}} = \text{verif}_T(y_i, y'_{\sigma^\star(i)})$$

By construction, minimal consistency pairs each completion with its most challenging counterpart.

**Random Baseline.** To calibrate these rewards, we evaluate their expected value when the model behaves as a random guesser. Identifying True $\mapsto 1$ and False $\mapsto 0$, we assume completions are independent unbiased bits:

$$y_i \overset{\text{iid}}{\sim} \text{Bernoulli}(1/2), \qquad y'_j \overset{\text{iid}}{\sim} \text{Bernoulli}(1/2),$$

with independence across the two groups.

---

**Proposition A.1: Expected Reward Under Random Baseline.**

Under the random baseline, the expected per-completion rewards for the three pairing strategies are:

(i) **Random pairing:** $\mathbb{E}[r_i^{\text{rand}}] = \frac{1}{2}$,

(ii) **One-to-all:** $\mathbb{E}[r_i^{\text{all}}] = \frac{1}{2}$,

(iii) **Minimal consistency:** $\mathbb{E}[r_i^{\text{OT}}] \approx \frac{1}{\sqrt{\pi K}}$.

In particular, only minimal consistency penalizes random guessing, with expected reward vanishing as $O(1/\sqrt{K})$.

---

*Proof.* *(i) Random pairing.* For each $i$, the pair $(y_i, y'_i)$ consists of two independent unbiased bits. Hence, $\mathbb{P}(y_i \neq y'_i) = \frac{1}{2}$. Since this holds for each $i$, we have $\mathbb{E}[r_i^{\text{rand}}] = \frac{1}{2}$.

*(ii) One-to-all.* The same argument applies to any pair $(y_i, y'_j)$: since all completions are independent, $\mathbb{P}(y_i \neq y'_j) = \frac{1}{2}$ for all $i, j$. Therefore $\mathbb{E}[r_i^{\text{all}}] = \frac{1}{2}$.

*(iii) Minimal consistency.* This case is more subtle. Define the counts:

$$S = \sum_{i=1}^{K} y_i, \qquad S' = \sum_{j=1}^{K} y'_j,$$

so that $S$ and $S'$ are the number of True in each group. Under the random baseline, $S, S' \sim \text{Bin}(K, 1/2)$ independently.

To minimize the number of disagreeing pairs, the optimal strategy matches identical bits whenever possible. Suppose $S \geq S'$ (the case $S' > S$ is symmetric). We match all $S'$ True values from the augmented group with True values from the original group (contributing 0 to the verifier), and similarly for False values. The remaining $S - S'$ True values must pair with False values, creating exactly $|S - S'|$ disagreements:

$$\min_{\sigma \in \mathcal{S}_K} \sum_{i=1}^{K} \mathbf{1}\{y_i \neq y'_{\sigma(i)}\} = |S - S'|.$$

The average reward is therefore $\frac{1}{K} \sum_i r_i^{\text{OT}} = |S - S'|/K$. To compute its expectation, we analyze $|S - S'|$. Write the difference as a sum of centered terms:

$$S - S' = \sum_{i=1}^{K} \left(y_i - \tfrac{1}{2}\right) - \sum_{j=1}^{K} \left(y'_j - \tfrac{1}{2}\right),$$

where each summand $(y_i - \frac{1}{2})$ or $(y'_j - \frac{1}{2})$ equals $\pm\frac{1}{2}$ with equal probability. This is a sum of $2K$ independent, mean-zero, bounded random variables. As $S, S' \sim \text{Bin}(K, 1/2)$ independently, the variance is:

$$\text{Var}(S - S') = K \cdot \frac{1}{4} + K \cdot \frac{1}{4} = \frac{K}{2}.$$

By the central limit theorem:

$$\frac{S - S'}{\sqrt{K/2}} \overset{d}{\to} \mathcal{N}(0, 1).$$

For a random variable $Z \sim \mathcal{N}(0, \sigma^2)$, we have $\mathbb{E}|Z| = \sigma\sqrt{2/\pi}$. Taking $\sigma = \sqrt{K/2}$ gives:

$$\mathbb{E}|S - S'| \approx \sqrt{\frac{K}{2}} \cdot \sqrt{\frac{2}{\pi}} = \sqrt{\frac{K}{\pi}}.$$

Therefore:

$$\mathbb{E}[r_i^{\mathrm{OT}}] = \frac{\mathbb{E}|S - S'|}{K} \approx \frac{1}{\sqrt{\pi K}}.$$

$\square$

Table 2 summarizes these results. Figure 9 visualizes this comparison. For typical group sizes ($K = 8$ to $32$), minimal consistency yields expected rewards between $0.20$ and $0.10$—well below the $0.50$ baseline of the other strategies.

*Table 2.* Expected per-completion reward under the random baseline (uninformative model).

| Strategy | $\mathbb{E}[r_i]$ | Scaling with $K$ |
|---|---|---|
| Random Pairing | $\mathbb{E}[r_i^{\mathrm{rand}}] = 1/2$ | Constant |
| One-to-All | $\mathbb{E}[r_i^{\mathrm{all}}] = 1/2$ | Constant |
| Minimal Consistency | $\mathbb{E}[r_i^{\mathrm{OT}}] \approx 1/\sqrt{\pi K}$ | $O(1/\sqrt{K})$ |

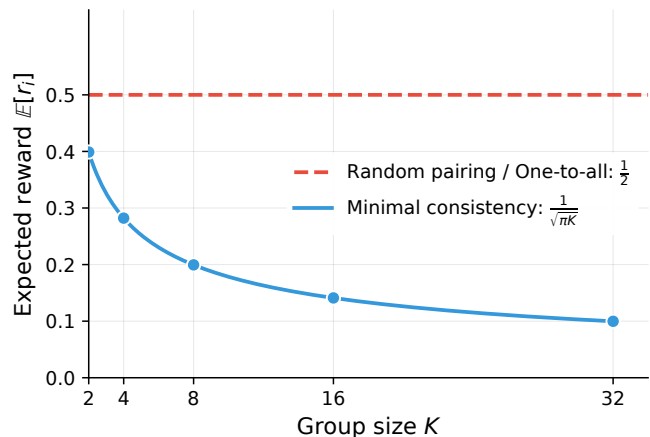

*Figure 9.* Expected per-completion reward under the random baseline as a function of group size $K$. Random pairing and one-to-all remain at $1/2$ regardless of $K$, while minimal consistency decays as $1/\sqrt{\pi K}$.

**Interpretation.** Only minimal consistency penalizes uninformative models: as $K$ grows, the expected reward for a random guesser vanishes as $O(1/\sqrt{K})$. The key insight is that minimal consistency actively searches for the *worst-case* pairing. A random model produces roughly equal numbers of True and False in each group, so the optimal matching pairs most completions with identical answers (which fail the verifier). Only the "leftover" completions—the imbalance $|S - S'| \sim \sqrt{K}$—contribute to the reward. In contrast, random pairing and one-to-all treat all pairs equally, allowing a random model to benefit from the $50\%$ chance that any given pair happens to satisfy the verifier.

**Conclusion.** Minimal consistency makes the consistency verification task intrinsically harder and the reward signal less susceptible to random guessing. This property is particularly valuable in our label-free setting, where we cannot rely on ground-truth accuracy to filter out uninformative behavior. By using minimal consistency, we ensure that high rewards reflect genuine cross-prompt agreement rather than statistical coincidence.

### A.2. Wasserstein Reformulation

Beyond the random-baseline analysis, minimal consistency admits an interpretation through optimal transport theory. We show that maximizing the minimal-consistency reward is equivalent to minimizing a Wasserstein distance between the model's output distributions on $x$ and $x'$, with transport cost given by the negative verifier $c_T = -\mathrm{verif}_T$.

---

**Definition A.2: Wasserstein Distance.**

Given two probability measures $\pi$ and $\pi'$ on a space $\mathcal{Y}$, and a cost function $c : \mathcal{Y} \times \mathcal{Y} \to \mathbb{R}$, the *Wasserstein distance* (or optimal transport cost) is:

$$W_c(\pi, \pi') := \min_{\gamma \in \Gamma(\pi, \pi')} \mathbb{E}_{(y, y') \sim \gamma} \big[ c(y, y') \big],$$

where $\Gamma(\pi, \pi')$ denotes the set of couplings—joint distributions on $\mathcal{Y} \times \mathcal{Y}$ with marginals $\pi$ and $\pi'$.

---

**Proposition A.3: Wasserstein Reformulation of OT-GRPO.**

Let $\pi_\theta^x := \pi_\theta(\cdot \mid x)$ and $\pi_\theta^{x'} := \pi_\theta(\cdot \mid x')$ denote the model's output distributions, and define the inconsistency cost $c_T := -\text{verif}_T$. Then the OT-GRPO objective is equivalent to minimizing the expected Wasserstein distance:

$$\min_\theta \; \mathbb{E}_{\substack{x \sim \mathcal{D}, \, T \sim \mathcal{T} \\ x' = T(x)}} \big[ W_{c_T}(\pi_\theta^x, \pi_\theta^{x'}) \big].$$

---

*Proof.* The initial pairwise consistency objective from Section 4 is:

$$\max_\theta \; \mathbb{E}_{\substack{x \sim \mathcal{D}, \, T \sim \mathcal{T} \\ x' = T(x)}} \mathbb{E}_{\substack{y \sim \pi_\theta(\cdot | x) \\ y' \sim \pi_\theta(\cdot | x')}} \big[ \text{verif}_T(y, y') \big].$$

In this formulation, $y$ and $y'$ are sampled independently. With minimal consistency pairing, however, completions are matched via the optimal transport coupling $\gamma^\star$ that minimizes expected consistency:

$$\gamma^\star = \arg\min_{\gamma \in \Gamma(\pi_\theta^x, \pi_\theta^{x'})} \mathbb{E}_{(y, y') \sim \gamma} \big[ \text{verif}_T(y, y') \big].$$

Substituting this adversarial coupling yields a max-min objective:

$$\max_\theta \; \mathbb{E}_{\substack{x \sim \mathcal{D}, \, T \sim \mathcal{T} \\ x' = T(x)}} \min_{\gamma \in \Gamma(\pi_\theta^x, \pi_\theta^{x'})} \mathbb{E}_{(y, y') \sim \gamma} \big[ \text{verif}_T(y, y') \big].$$

Recognizing the inner minimization as the Wasserstein distance $W_{\text{verif}_T}(\pi_\theta^x, \pi_\theta^{x'})$:

$$\max_\theta \; \mathbb{E}_{\substack{x \sim \mathcal{D}, \, T \sim \mathcal{T} \\ x' = T(x)}} W_{\text{verif}_T}(\pi_\theta^x, \pi_\theta^{x'}).$$

Converting maximization to minimization by negating:

$$\min_\theta \; \mathbb{E}_{\substack{x \sim \mathcal{D}, \, T \sim \mathcal{T} \\ x' = T(x)}} \big( - W_{\text{verif}_T}(\pi_\theta^x, \pi_\theta^{x'}) \big).$$

Finally, absorbing the negative into the cost gives $-W_{\text{verif}_T} = W_{-\text{verif}_T} = W_{c_T}$, completing the proof. $\qquad \square$

**Interpretation.** This reformulation reveals that OT-GRPO seeks to *align* the model's conditional distributions $\pi_\theta^x$ and $\pi_\theta^{x'}$ in the geometry defined by the Wasserstein distance with inconsistency cost $c_T = -\text{verif}_T$.

## B. Implementation Details

### B.1. System Prompt

As described in Section 4, we prompt the model with a system instruction adapted from DeepSeek-R1 (DeepSeek-AI, 2025) to encourage chain-of-thought reasoning. The exact system prompt is:

> *"A conversation between User and Assistant. The user asks a question, and the Assistant solves it. The assistant first thinks about the reasoning process in the mind and then provides the user with the answer. The reasoning process and answer are enclosed within* `<think> </think>` *and* `<answer> </answer>` *tags, respectively, i.e.,* `<think>` *reasoning process here* `</think><answer>` *answer here* `</answer>`."*

The format reward is 1 if the completion contains both a valid <think>...</think> block and a valid <answer>...</answer> block with the answer being either "True" or "False". If either component is missing or malformed, the format reward is 0.

## B.2. Handling Unparseable Answers

During training, some completions may fail to produce a valid "True" or "False" answer (e.g., due to malformed output or refusal to answer). We handle these *unparseable* completions as follows:

- Unparseable completions are **excluded from the OT matching**. The optimal transport problem is solved only over completions with valid parsed answers.

- Unparseable completions receive a **consistency reward of 0**, which—after group normalization—translates to a negative advantage, discouraging the model from producing unparseable outputs.

This design ensures that the OT matching remains well-defined even when some completions are invalid, while providing a learning signal that encourages properly formatted outputs. The formal treatment of the matching problem—including how we handle unequal group sizes—is given in Section B.3.

## B.3. Discrete Optimal Transport for Completion Matching

This section details how we solve the minimal consistency matching in practice, particularly when unparseable completions lead to unequal group sizes. As discussed in Section B.2, unparseable completions are excluded, leaving $n_0$ valid completions $\{y_1, \ldots, y_{n_0}\}$ from the original prompt and $n_1$ valid completions $\{y'_1, \ldots, y'_{n_1}\}$ from the augmented prompt.

---

**Definition B.1: Discrete Optimal Transport for Completion Matching.**

Given $n_0$ valid completions $\{y_1, \ldots, y_{n_0}\}$ from the original prompt and $n_1$ valid completions $\{y'_1, \ldots, y'_{n_1}\}$ from the augmented prompt, we seek a coupling matrix $\gamma \in \mathbb{R}^{n_0 \times n_1}$ that minimizes the total consistency cost:

$$\gamma^\star = \arg\min_{\gamma \geq 0} \sum_{i=1}^{n_0} \sum_{j=1}^{n_1} \gamma_{ij} \cdot \mathrm{verif}_T(y_i, y'_j) \quad \text{s.t.} \quad \sum_{j=1}^{n_1} \gamma_{ij} = \frac{1}{n_0}, \quad \sum_{i=1}^{n_0} \gamma_{ij} = \frac{1}{n_1}.$$

The constraints ensure mass conservation, namely, that each original completion has total outgoing mass $1/n_0$ and each augmented completion has total incoming mass $1/n_1$.

---

**Equal group sizes ($n_0 = n_1 = K$).** When all completions are parseable, the feasible set is the *Birkhoff polytope*—the set of doubly stochastic matrices (scaled by $1/K$). By the Birkhoff–von Neumann theorem (Birkhoff, 1946), its vertices are precisely the permutation matrices. Since we minimize a linear cost, the optimum $\gamma^\star$ is attained at a vertex, corresponding to a permutation $\sigma^\star \in \mathcal{S}_K$. This recovers the formulation in Section 4.

**Unequal group sizes ($n_0 \neq n_1$).** When some completions are unparseable (by not following the desired format), the marginal constraints differ and the feasible set is no longer the Birkhoff polytope. The optimal coupling $\gamma^\star$ may assign fractional mass from one completion to multiple partners.

**Extracting deterministic assignments.** To obtain per-completion rewards, we extract a deterministic assignment from $\gamma^\star$ via argmax:

$$\text{For } y_i: \quad j^\star(i) = \arg\max_{j \in \{1, \ldots, n_1\}} \gamma^\star_{ij}, \qquad \text{For } y'_j: \quad i^\star(j) = \arg\max_{i \in \{1, \ldots, n_0\}} \gamma^\star_{ij}.$$

The reward for $y_i$ is $\mathrm{verif}_T(y_i, y'_{j^\star(i)})$, and similarly for $y'_j$. When $n_0 = n_1$, the optimal coupling is a permutation and argmax recovers the unique partner. When $n_0 \neq n_1$, argmax selects the partner receiving the largest transport mass.

This formulation gracefully handles missing data: with all completions parseable, we recover the permutation matching; with some unparseable, we solve the unbalanced OT problem and extract assignments accordingly.

## C. Qualitative Example

Figure 10 shows the model's reasoning before and after consistency training on a depth comparison task from SUN RGB-D. Before training, the model relies on a flawed heuristic (vertical position in the image) and produces an incorrect answer. After training, the model correctly reasons about 3D spatial relationships and arrives at the correct answer.

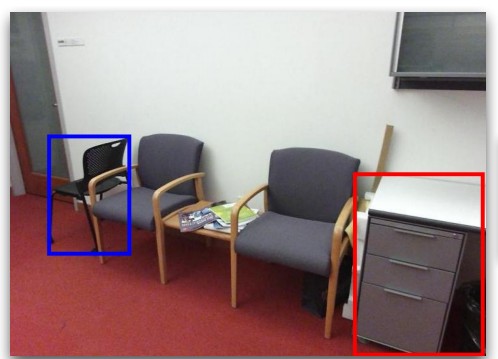

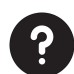

- **object 1** = "**desk**", highlighted by a **red** box.
- **object 2** = "**chair**", highlighted by a **blue** box.
Is object 1 **closer** to the camera than object 2? True/False.

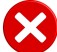

**Before Training**

<think>
To answer this question, I need to analyze the 3D spatial arrangement of the desk and chair relative to the camera position.

The chair with the blue bounding box is positioned in the left foreground of the scene. I can see its legs clearly extending toward the viewer, and it sits in the near field of the room. The desk with the red bounding box is actually a filing cabinet positioned further back in the room, closer to the back wall.

Looking at the depth ordering, the chair occupies the foreground space while the filing cabinet is set back in the mid-ground near the wall. The chair's proximity to the camera is evident from its larger apparent size relative to its actual dimensions and its position in the scene's depth layers.

Therefore, the desk is not closer to the camera—the chair is closer.
</think>
<answer>False</answer>

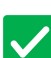

**After Training**

<think>
I need to determine which object is closer to the camera by analyzing their positions in the image. The desk is marked with a red bounding box on the right side of the frame, and the chair is marked with a blue bounding box on the left side.

Looking at the visual cues, objects that appear lower in the image plane are typically closer to the camera due to perspective projection. The desk seems to be positioned at a lower vertical position in the frame. Additionally, the desk occupies a larger portion of the image area, which suggest it's closer to the viewer.

The chair appears to be elevated and positioned more toward the background of the scene. Based on these observations about vertical position and apparent size, I conclude that the desk is likely closer to the camera than the chair.
</think>
<answer>True</answer>

*Figure 10.* Qualitative example showing how consistency training improves spatial reasoning. Given a depth comparison question from SUN RGB-D, the Qwen2.5-VL-7B model initially produces an incorrect answer based on flawed heuristics (left), while after training, the model reasons correctly about 3D spatial relationships and arrives at the correct answer (right).

## D. Dataset Construction

Spatial reasoning tasks require unambiguous ground-truth answers to train and evaluate models effectively. We construct our datasets from images with 3D object annotations, applying careful filtering to select object pairs with clear spatial relationships. This section describes the data sources, task definitions, selection criteria, and transformations used to generate training examples.

### D.1. Source Data

**Omni3D Annotations.** Spatial reasoning tasks require ground-truth 3D information to establish unambiguous answers. We use annotations from Omni3D (Brazil et al., 2023), a large-scale benchmark that provides unified 3D bounding boxes across multiple RGB datasets. Each annotation includes a 9-DoF 3D bounding box (position, orientation, dimensions) in camera coordinates, along with 2D projections and visibility estimates. All 3D quantities are expressed in camera coordinates, where the x-axis points right, the y-axis points down, and the z-axis points forward into the scene. Table 3 compares the two source datasets.

**KITTI (Outdoor).** We use KITTI (Geiger et al., 2012), an autonomous driving benchmark with forward-facing stereo images captured from a moving vehicle. Objects appear at depths from 5 to 100 meters, and the clear lane structure provides unambiguous left/right relationships.

*Table 3.* Comparison of source datasets. KITTI provides outdoor driving scenes with large depth ranges; SUN RGB-D provides cluttered indoor scenes with diverse object arrangements.

| Property | KITTI | SUN RGB-D |
|---|---|---|
| Domain | Outdoor (driving) | Indoor (rooms) |
| Camera setup | Forward-facing stereo | RGB-D sensors |
| Depth range | 5–100 m | 0.5–10 m |
| Typical objects | Cars, pedestrians, cyclists | Furniture, appliances |
| Scene structure | Lane-based, sparse | Cluttered, dense |

**SUN RGB-D (Indoor).** We use SUN RGB-D (Song et al., 2015), a collection of RGB-D data from diverse indoor environments. Objects appear at closer range (0.5–10 m), scenes are more cluttered with frequent occlusions, and spatial arrangements are more varied. The contrast between domains allows us to evaluate cross-domain generalization.

### D.2. Task Definitions

We define four spatial reasoning tasks, each formulated as a binary True/False question. All tasks follow the same structure: given an image with highlighted objects, the model must determine whether a stated spatial relationship holds. The tasks are designed to cover different aspects of 3D spatial understanding: horizontal position (orientation), camera-relative depth, inter-object distance, and physical size.

*Table 4.* Summary of spatial reasoning tasks. Each task is a binary True/False question about the spatial relationship between highlighted objects.

| Task | Example Question | Ground Truth | Marker |
|---|---|---|---|
| Orientation | Is A left of B? | 2D bbox center $x$ | Dots |
| Depth | Is A closer than B? | Min $z$ of 3D bbox | Boxes |
| Size | Is A bigger than B? | 3D volume | Dots |
| Rel. Distance | Is A closer to C than B? | 3D Euclidean dist. | Boxes |

*Table 5.* Dataset statistics. Number of training examples per task and source dataset, after filtering. Tasks may share images, so counts should not be summed across tasks.

| Task | KITTI | SUN RGB-D |
|---|---|---|
| Orientation | 2,406 | 7,738 |
| Depth | 4,288 | 6,222 |
| Size | 3,996 | 4,845 |
| Rel. Distance | 3,539 | 6,893 |

**Orientation.** Ground truth is determined by comparing the x-coordinates of 2D bounding box centers. We use colored dots as markers to avoid revealing positional information through bounding box placement.

**Depth.** Ground truth is the minimum z-coordinate across all eight 3D bounding box corners (using the minimum rather than the center handles objects extending toward the camera). We use bounding boxes as markers, since 2D box size correlates only weakly with depth.

**Size.** Ground truth is 3D volume (width $\times$ height $\times$ length). We use dots as markers to avoid revealing size through bounding box dimensions.

**Relative Distance.** This task involves three objects: an anchor and two comparison objects. Ground truth is the Euclidean distance between object centers. We select the anchor to maximize the distance gap between comparison objects.

**Linguistic Variation.** For each task, we define five question templates that paraphrase the same underlying question and two relation phrases (e.g., "left of" / "right of"). During training, templates and relations are sampled to encourage learning the spatial concept rather than pattern-matching specific phrasings. Table 6 lists all templates.

*Table 6.* Question templates and relation phrases for each task. `{IDX0}` and `{IDX1}` are placeholders for object indices; `{REL}` is replaced by one of the relation phrases listed.

| Task | Relations | Templates |
|---|---|---|
| **Orientation** | • `left of`
• `right of` | • `Is object {IDX0} to the {REL} object {IDX1}?`
• `In the image, is object {IDX0} {REL} object {IDX1}?`
• `Compared to object {IDX1}, is object {IDX0} to the {REL} it?`
• `Looking at the scene, is object {IDX0} {REL} object {IDX1}?`
• `Between the two objects, is object {IDX0} to the {REL} object {IDX1}?` |
| **Depth** | • `closer to`
• `further from` | • `Is object {IDX0} {REL} the camera than object {IDX1}?`
• `In terms of depth, is object {IDX0} {REL} the camera than object {IDX1}?`
• `Compared to object {IDX1}, is object {IDX0} {REL} the camera?`
• `Does object {IDX0} appear {REL} the camera than object {IDX1}?`
• `Between the two objects, is object {IDX0} {REL} the camera than object {IDX1}?` |
| **Size** | • `bigger than`
• `smaller than` | • `Is object {IDX0} {REL} object {IDX1}?`
• `In terms of size, is object {IDX0} {REL} object {IDX1}?`
• `Compared to object {IDX1}, is object {IDX0} {REL}?`
• `Does object {IDX0} appear {REL} object {IDX1}?`
• `Between the two objects, is object {IDX0} {REL} object {IDX1}?` |
| **Rel. Distance** | • `closer to`
• `further from` | • `Is object {IDX0} {REL} object 1 than object {IDX1}?`
• `In 3D space, is object {IDX0} {REL} object 1 than object {IDX1}?`
• `Compared to object {IDX1}, is object {IDX0} {REL} object 1?`
• `Looking at the scene, is object {IDX0} {REL} object 1 than object {IDX1}?` |

**Answer Balance.** Because we sample relation phrases and object orders uniformly at random, the ground-truth answers are balanced: approximately 50% True and 50% False for each task in both datasets.

### D.3. Object and Pair Selection

Not all object pairs yield useful training examples. Heavily occluded objects may be difficult to identify; objects with nearly identical depths create ambiguous comparisons; overlapping bounding boxes confuse left/right judgments. We apply a series of filters to select high-quality examples.

**Single-Object Filters.** Each object must satisfy validity criteria before being considered for pairing. Table 7 summarizes the thresholds. We require sufficient visibility (using the Omni3D visibility score), reasonable 2D bounding box size (large enough to identify, small enough not to dominate the frame), and—for KITTI—that the 3D center lies in front of the camera plane.

**Pair and Group Filters.** When two objects overlap significantly, spatial relationships become ambiguous. We filter pairs by bounding box IoU (intersection over union) and coverage (fraction of one box contained in another). Stricter thresholds

*Table 7.* Single-object filter thresholds. Objects failing any criterion are excluded.

| Filter | KITTI | SUN RGB-D |
|---|---|---|
| Min visibility | 15% | 10% |
| Min area (image fraction) | 0.5% | 2% |
| Max area (image fraction) | 70–80% | 80% |
| Reject behind camera | Yes | No |

apply to same-class pairs, where visual confusion is more likely. Each task also requires sufficient separation along its relevant dimension. Table 8 lists the thresholds.

*Table 8.* Pair/group filter thresholds. Pairs exceeding overlap limits or failing to meet minimum gaps are excluded.

| Filter | KITTI | SUN RGB-D |
|---|---|---|
| Max IoU (any class) | 35% | 30% |
| Max IoU (same class) | 15% | 10% |
| Max coverage (any class) | 35% | 30% |
| Max coverage (same class) | 15% | 10% |
| Min depth gap (Depth task) | 0.5 m | 0.3 m |
| Min volume gap (Size task) | $1.0\,\mathrm{m}^3$ | $0.5\,\mathrm{m}^3$ |
| Min distance gap (Rel. Dist.) | 0.5 m | 0.3 m |

**Selection Strategy.** After filtering, multiple valid pairs typically remain per image. We score each pair based on the metric gap and object size, then select according to a task-specific strategy.

### D.4. Transformations for Consistency Training

The consistency training approach described in Section 4 requires paired prompts where the relationship between correct answers is known without access to ground-truth labels. We achieve this through transformations applied to both images and questions, each with a known effect on the correct answer.

**Image Transformations.** We apply three types of image transformations. Horizontal flips mirror the image left-to-right, swapping the horizontal positions of all objects; we update the stored 2D and 3D bounding box coordinates accordingly. Bounding-box-preserving crops randomly select a sub-region of the image while ensuring all annotated objects remain fully visible; the crop scale ranges from 70% to 100% of the original image. Color adjustments (brightness, contrast, saturation) modify the image appearance without affecting spatial relationships.

**Text Transformations.** We vary the question formulation through three mechanisms. Template sampling selects among the five paraphrased question templates for each task. Relation swapping replaces the queried relation with its opposite (e.g., "closer to" becomes "further from"). Object order swapping permutes which object is referred to as "object 1" versus "object 2" in the question. Each mechanism changes the surface form of the question while preserving or predictably altering its meaning.

**Invariant Transformations.** Some transformations preserve the correct answer. Color adjustments do not affect spatial relationships, so the answer remains unchanged. Bounding-box-preserving crops maintain all objects and their relative positions, leaving the answer unchanged. These invariant transformations create prompt pairs where the consistency verifier expects matching answers.

**Equivariant Transformations.** Other transformations predictably change the correct answer. A horizontal flip swaps left and right, negating the answer to orientation questions—if A was left of B before the flip, A is right of B after. Relation swapping negates the answer by asking the opposite question. Object order swapping negates the answer for symmetric relations: "Is A closer than B?" and "Is B closer than A?" have opposite answers. These equivariant transformations create prompt pairs where the verifier expects opposite answers. Table 9 summarizes the transformation properties.

*Table 9.* Transformation properties. Invariant transformations preserve the answer; equivariant transformations negate it.

| Transformation | Type | Effect on Answer |
|---|---|---|
| Color adjustments | Invariant | No change |
| Bbox-preserving crop | Invariant | No change |
| Horizontal flip | Equivariant | Negates orientation |
| Relation swap | Equivariant | Negates answer |
| Object order swap | Equivariant | Negates answer |

**Composition Rule.**   When multiple transformations are applied simultaneously, their effects compose according to a simple rule: each equivariant transformation contributes one negation, and an even number of negations cancels out. For example, applying both a horizontal flip and a relation swap results in two negations, so the final answer matches the original. Applying a flip, a relation swap, and an object order swap results in three negations, so the final answer is negated. This composition rule allows the consistency verifier to determine the expected answer relationship for any combination of transformations.

**Sampling Strategy.**   In practice, we use all transformations for all tasks. Each transformation is applied independently with probability 0.5, and the sampled transformations are composed. This stochastic composition ensures diverse augmentation during training while maintaining the ability to compute the expected answer relationship via the composition rule.

With the filtered object pairs, question templates, and transformation definitions in place, we have all the building blocks needed to generate paired prompts at training time. During each training step, transformations are sampled and applied on-the-fly to create the original and augmented prompt pairs used by the consistency verifier.

### D.5. Example Prompt Pairs

Figures 11–14 show example prompt pairs for each of the four spatial reasoning tasks. Each figure displays an original prompt (left) alongside an augmented prompt (right) created by applying various image and text transformations. The examples demonstrate different transformation combinations: some use only invariant transforms (where answers should match), while others include equivariant transforms that predictably change the answer. The prompts show the object descriptions and questions presented to the model; the colored markers (boxes or dots) indicate the referenced objects.

### D.6. KITTI Example Prompt Pairs

Figures 15–18 show example prompt pairs from the KITTI dataset (outdoor driving scenes). KITTI images feature sparser layouts with objects at greater depths (5–100 m), predominantly containing vehicles (cars, vans, trucks) and pedestrians. The same transformation strategies apply as in SUN RGB-D.

### D.7. Numeric Task Example Prompt Pairs

Figures 19–20 show example prompt pairs for the two numeric tasks introduced in Section 5. As discussed there, the answer (an integer count or a metric distance) is invariant under every transformation we apply, so the consistency verifier expects the original and augmented predictions to be *equal*—there is no equivariant case to consider.

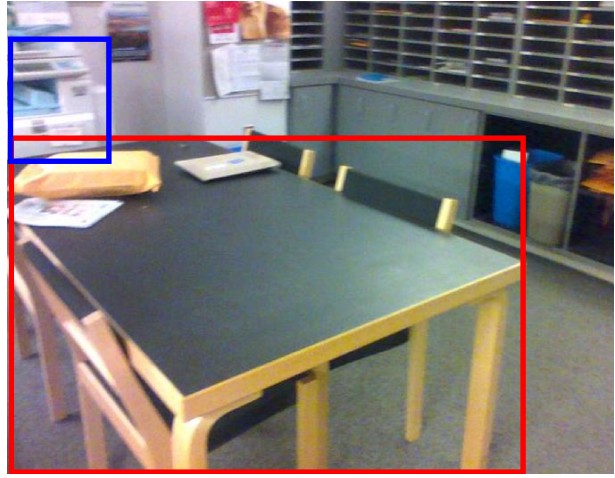 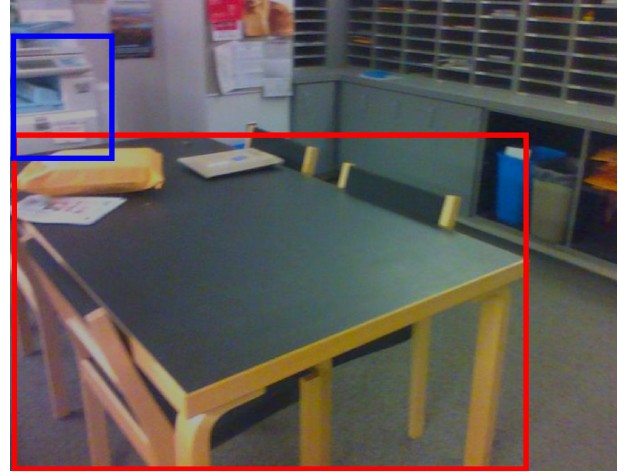

```
- object 1 = "table", highlighted by a red
box.
- object 2 = "printer", highlighted by a
blue box.
Is object 1 closer to the camera than
object 2?
```
**Answer: True**

```
- object 1 = "table", highlighted by a red
box.
- object 2 = "printer", highlighted by a
blue box.
Is object 1 further from the camera than
object 2?
```
**Answer: False**

*Figure 11.* Depth task example. The augmented prompt applies color jitter and relation swap ("closer to" → "further from"), but no horizontal flip. Since relation swap is a single equivariant transformation (one negation), the **answers should differ**.

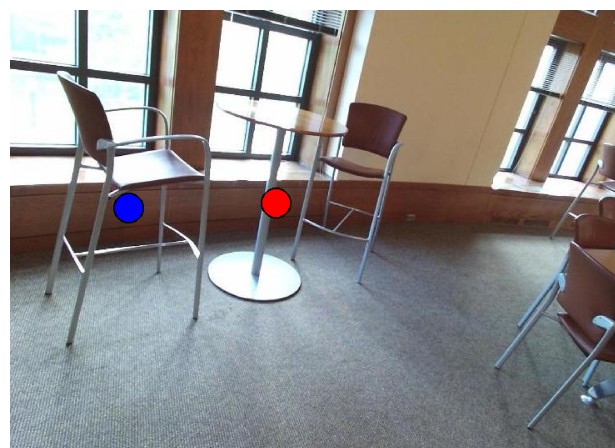 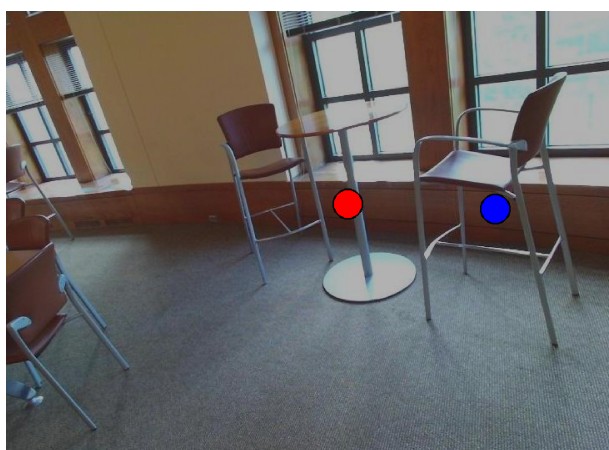

```
- object 1 = "table", marked with a red
dot.
- object 2 = "chair", marked with a blue
dot.
Is object 1 to the left of object 2?
```
**Answer: False**

```
- object 1 = "table", marked with a red
dot.
- object 2 = "chair", marked with a blue
dot.
Is object 1 to the right of object 2?
```
**Answer: False**

*Figure 12.* Orientation task example. The augmented prompt applies horizontal flip, color jitter, and relation swap ("left of" → "right of"). The flip negates the spatial relationship, and the relation swap negates the question—two equivariant transformations that cancel out, so the **answers should match**.

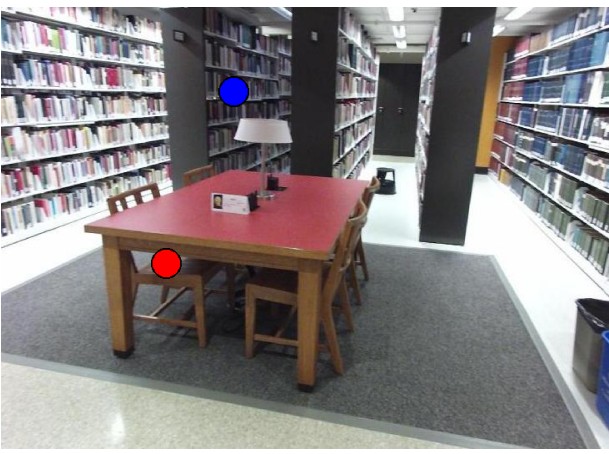 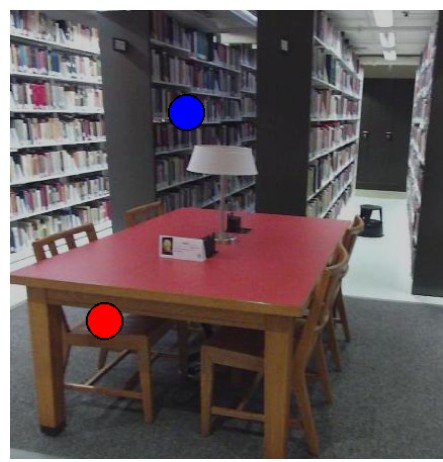

```
- object 1 = "chair", marked with a red
dot.
- object 2 = "bookcase", marked with a blue
dot.
Is object 1 bigger than object 2?
```
**Answer: False**

```
- object 1 = "chair", marked with a red
dot.
- object 2 = "bookcase", marked with a blue
dot.
Compared to object 2, is object 1 bigger
than?
```
**Answer: False**

*Figure 13.* Size task example. The augmented prompt applies an aggressive crop (50–60% scale), color jitter, and a different question template—but no relation swap. All transformations are invariant (zero negations), so the **answers should match**.

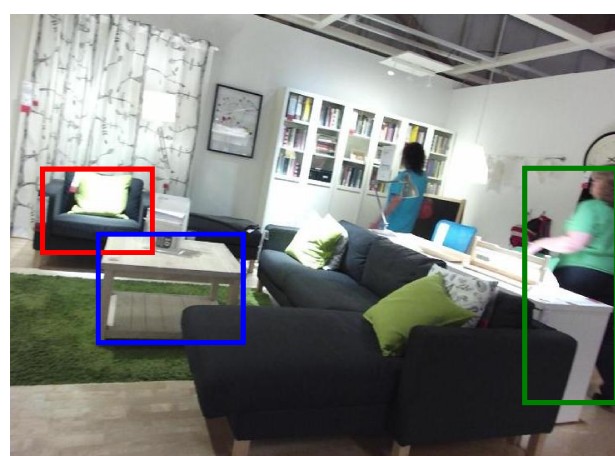 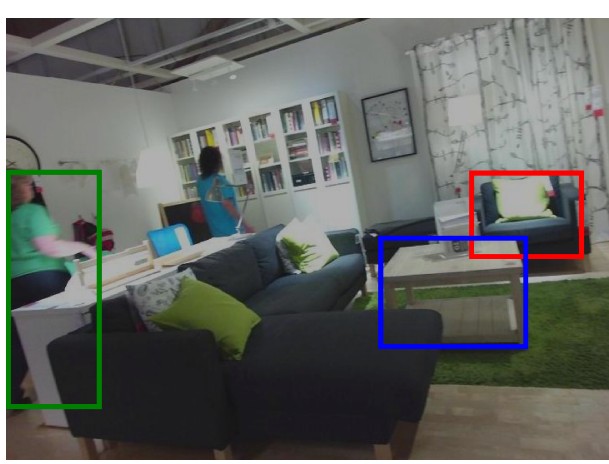

```
- object 1 = "chair", highlighted by a red
box.
- object 2 = "table", highlighted by a blue
box.
- object 3 = "person", highlighted by a
green box.
Is object 2 closer to object 1 than object
3?
```
**Answer: True**

```
- object 1 = "chair", highlighted by a red
box.
- object 2 = "table", highlighted by a blue
box.
- object 3 = "person", highlighted by a
green box.
Is object 2 further from object 1 than
object 3?
```
**Answer: False**

*Figure 14.* Relative distance task example (triplet). The augmented prompt applies horizontal flip, color jitter, and relation swap ("closer to" → "further from"). The flip does not affect inter-object distances, so only the relation swap contributes a negation—the **answers should differ**.

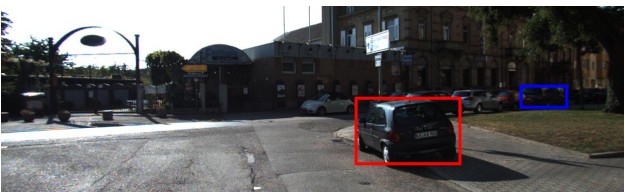 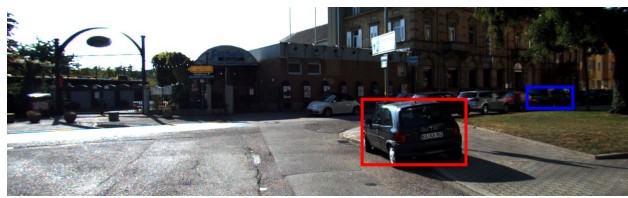

```
- object 1 = "car", highlighted by a red
box.
- object 2 = "car", highlighted by a blue
box.
Is object 1 closer to the camera than
object 2?
```
**Answer: True**

```
- object 1 = "car", highlighted by a red
box.
- object 2 = "car", highlighted by a blue
box.
Is object 1 further from the camera than
object 2?
```
**Answer: False**

*Figure 15.* KITTI depth task example. The augmented prompt applies color jitter and relation swap ("closer to" → "further from"), but no horizontal flip. Since relation swap is a single equivariant transformation (one negation), the **answers should differ**.

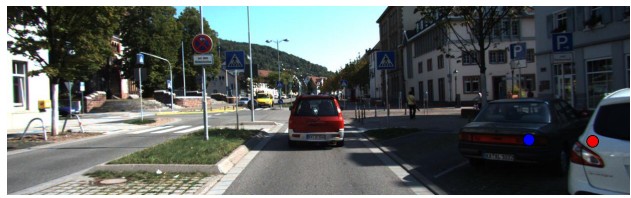 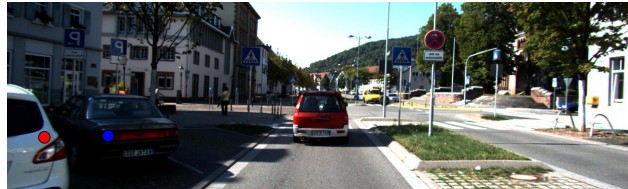

```
- object 1 = "car", marked with a red dot.
- object 2 = "car", marked with a blue dot.
Is object 1 to the left of object 2?
```
**Answer: False**

```
- object 1 = "car", marked with a red dot.
- object 2 = "car", marked with a blue dot.
Is object 1 to the right of object 2?
```
**Answer: False**

*Figure 16.* KITTI orientation task example. The augmented prompt applies horizontal flip, color jitter, and relation swap ("left of" → "right of"). The flip negates the spatial relationship, and the relation swap negates the question—two equivariant transformations that cancel out, so the **answers should match**.

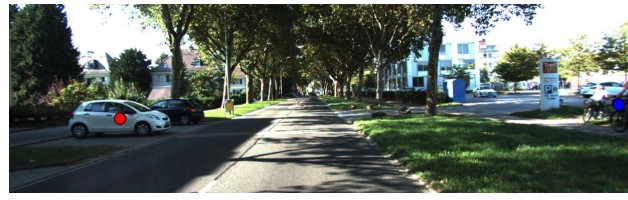 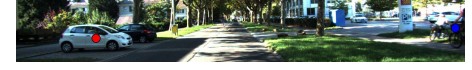

```
- object 1 = "car", marked with a red dot.
- object 2 = "cyclist", marked with a blue
dot.
Is object 1 bigger than object 2?
```
**Answer: False**

```
- object 1 = "car", marked with a red dot.
- object 2 = "cyclist", marked with a blue
dot.
Compared to object 2, is object 1 larger?
```
**Answer: False**

*Figure 17.* KITTI size task example. The augmented prompt applies a bounding-box-aware crop, color jitter, and a different question template—but no relation swap. All transformations are invariant (zero negations), so the **answers should match**.

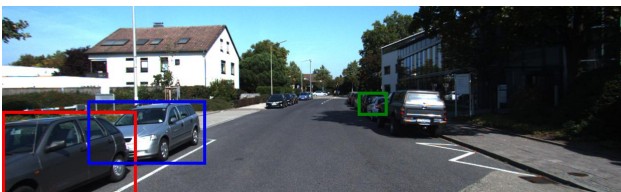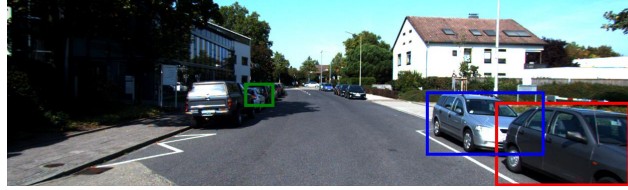

```
- object 1 = "car", highlighted by a red
box.
- object 2 = "car", highlighted by a blue
box.
- object 3 = "car", highlighted by a green
box.
Is object 2 closer to object 1 than object
3?
```
**Answer: True**

```
- object 1 = "car", highlighted by a red
box.
- object 2 = "car", highlighted by a blue
box.
- object 3 = "car", highlighted by a green
box.
Is object 2 further from object 1 than
object 3?
```
**Answer: False**

*Figure 18.* KITTI relative distance task example (triplet). The augmented prompt applies horizontal flip, color jitter, and relation swap ("closer to" → "further from"). The flip does not affect inter-object distances, so only the relation swap contributes a negation—the **answers should differ**.

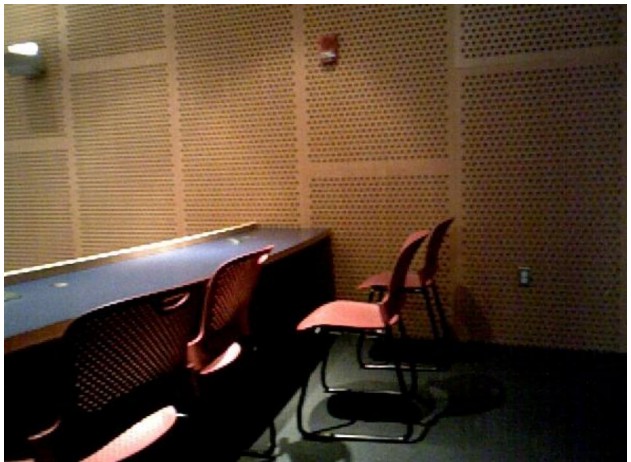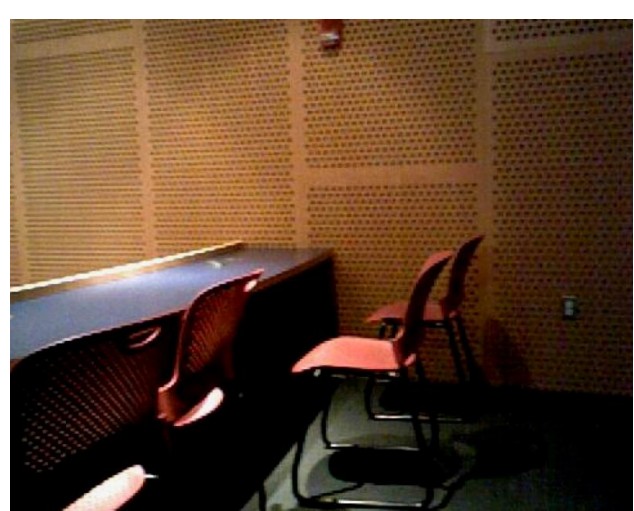

```
How many chairs are visible in the image?
```
**Answer: 3**

```
In the image, how many chairs can you see?
```
**Answer: 3**

*Figure 19.* Counting task example. The augmented prompt applies an object-preserving crop, color jitter, and template resampling. All transformations are invariant for counting (the number of objects of a given class is unchanged), so the **answers should match**.

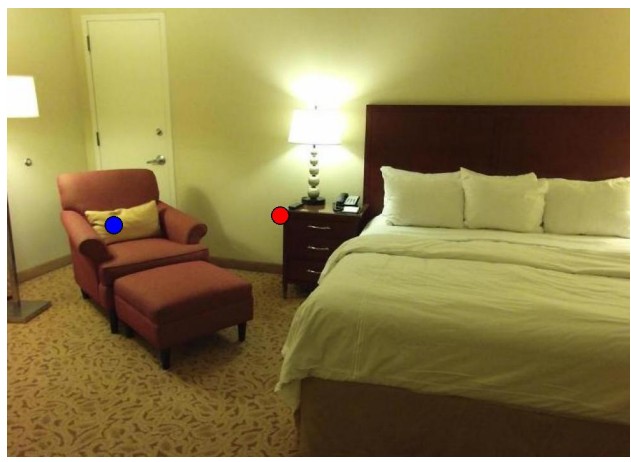 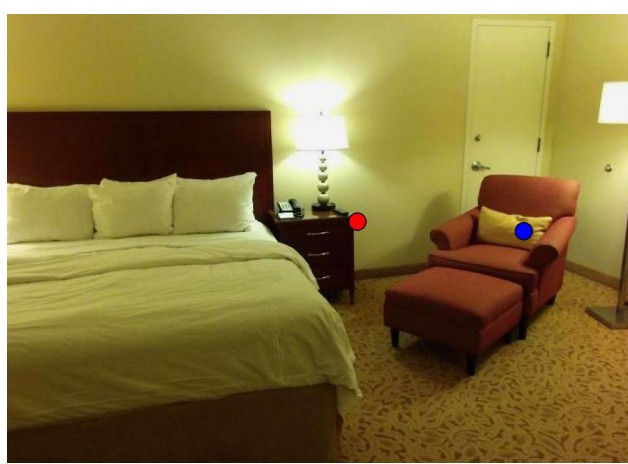

```
- object 1 = "table", marked with a red
dot.
- object 2 = "chair", marked with a blue
dot.
What is the distance between object 1 and
object 2 in meters?
```
**Answer: 1.4**

```
- object 1 = "table", marked with a red
dot.
- object 2 = "chair", marked with a blue
dot.
In meters, how far apart are object 1 and
object 2?
```
**Answer: 1.4**

*Figure 20.* Absolute distance task example. The augmented prompt applies horizontal flip, color jitter, and template resampling. Mirroring and visual jitter leave the 3D distance between two objects unchanged and the question paraphrasing keeps its meaning, so the **answers should match**.

