# A. Analysis of Consistency Pairing Strategies

This appendix extends the analysis of pairing strategies introduced in Section 4, providing a more detailed interpretation of why minimal consistency pairing may be preferable to simpler alternatives like random pairing or one-to-all. We investigate how different pairing strategies behave when applied to an uninformative model—one that guesses uniformly at random, ignoring the transformation relationship between prompts. By deriving the expected per-completion reward $\mathbb{E}[r_i]$ under this random baseline, we can compare how permissive each strategy is: strategies that assign high reward to random guessing are easier to "hack," while strategies that assign low reward provide a stronger learning signal.

## A.1. Setup and Notation

Let $x$ be an original prompt and $x' = T(x)$ an augmented version obtained by applying a transformation $T$. We sample $K$ binary completions from each prompt:

$$y_1, \ldots, y_K \sim \pi_\theta(\cdot \mid x), \qquad y'_1, \ldots, y'_K \sim \pi_\theta(\cdot \mid x'),$$

where each completion $y_i, y'_j \in \{\text{True}, \text{False}\}$.

As described in Section 3, each transformation $T$ induces a known mapping $\phi_T$ on the answer space: if $y^\star$ is the correct answer to $x$, then $\phi_T(y^\star)$ is the correct answer to $x'$. For binary tasks, $\phi_T$ is either the identity (invariant transformations, where answers should match) or negation (equivariant transformations, where answers should differ). The consistency verifier rewards completions that satisfy the expected relationship:

$$\text{verif}_T(y, y') = \mathbf{1}\{y' = \phi_T(y)\} = \begin{cases} \mathbf{1}\{y = y'\} & \text{if } \phi_T = \text{id (invariant)}, \\ \mathbf{1}\{y \neq y'\} & \text{if } \phi_T = \neg \text{ (equivariant)}. \end{cases}$$

For the random baseline analysis below, we consider the equivariant case $\text{verif}_T(y, y') = \mathbf{1}\{y \neq y'\}$ without loss of generality. The invariant case $\text{verif}_T(y, y') = \mathbf{1}\{y = y'\}$ is symmetric: under the random baseline where completions are independent unbiased bits, $\mathbb{P}(y = y') = \mathbb{P}(y \neq y') = 1/2$, so all expected values derived below remain unchanged.

## A.2. Pairing Strategies and Per-Completion Rewards

Given $K$ completions from each prompt, we have a $K \times K$ matrix of verifier scores $V_{ij} = \text{verif}_T(y_i, y'_j)$. Different strategies aggregate this matrix into per-completion rewards $r_i$.

**Random Pairing.** Pair completions by generation order—the $i$-th completion from $x$ with the $i$-th completion from $x'$. The per-completion reward is:
$$r_i^{\text{rand}} = \text{verif}_T(y_i, y'_i).$$

This strategy is simple but arbitrary: the reward depends on the sampling order rather than the content of the completions.

**One-to-All.** Compare each completion from $x$ to every completion from $x'$ and average:

$$r_i^{\text{all}} = \frac{1}{K} \sum_{j=1}^{K} \text{verif}_T(y_i, y'_j).$$

Averaging over all pairs reduces variance but can dilute the signal: a few strong inconsistencies may be hidden when most pairs happen to be consistent.

**Minimal Consistency (OT-GRPO).** Find the one-to-one matching $\sigma^\star \in \mathcal{S}_K$ (a permutation) that *minimizes* the total consistency score:

$$\sigma^\star \in \arg\min_{\sigma \in \mathcal{S}_K} \sum_{i=1}^{K} \text{verif}_T(y_i, y'_{\sigma(i)}).$$

The per-completion reward is: $r_i^{\text{OT}} = \text{verif}_T(y_i, y'_{\sigma^\star(i)})$. By construction, minimal consistency pairs each completion with its most challenging counterpart—the one that creates the smallest reward.

### A.3. Random Baseline Analysis

To calibrate these rewards, we evaluate their expected value under an *uninformative* regime where the model carries no signal about the transformation and behaves like a random guesser. For simplicity, we identify $\text{True} \mapsto 1$ and $\text{False} \mapsto 0$, so completions become binary values. Concretely, we assume completions are independent unbiased bits:

$$y_i \overset{\text{iid}}{\sim} \text{Bernoulli}(1/2), \qquad y_j' \overset{\text{iid}}{\sim} \text{Bernoulli}(1/2),$$

with independence across the two groups. Rewards that remain high under this null are more permissive (easier to "hack"), while rewards that drop are more conservative.

**Random Pairing.** For each $i$, the pair $(y_i, y_i')$ consists of two independent unbiased bits. Hence, $\mathbb{P}(y_i \neq y_i') = \frac{1}{2}$. Since this holds for each $i$, we have $\mathbb{E}[r_i^{\text{rand}}] = \frac{1}{2}$.

**One-to-All.** The same argument applies to any pair $(y_i, y_j')$: since all completions are independent, $\mathbb{P}(y_i \neq y_j') = \frac{1}{2}$ for all $i, j$. Therefore $\mathbb{E}[r_i^{\text{all}}] = \frac{1}{2}$.

**Minimal Consistency.** This case is more subtle. Define the counts:

$$S = \sum_{i=1}^{K} y_i, \qquad S' = \sum_{j=1}^{K} y_j',$$

so that $S$ and $S'$ are the number of True in each group. Under the random baseline, $S, S' \sim \text{Bin}(K, 1/2)$ independently.

To minimize the number of disagreeing pairs, we should match identical bits whenever possible. Suppose $S \geq S'$ (the case $S' > S$ is symmetric). We can match all $S'$ True values from the augmented group with $S'$ True values from the original group—these pairs agree (both True), contributing 0 to the verifier sum. Similarly, we match as many False values as possible. After these matches, $S - S'$ True values in the original group remain unmatched with True values; they must be paired with False values, creating exactly $S - S'$ disagreements. No matching can achieve fewer disagreements, so:

$$\min_{\sigma \in \mathcal{S}_K} \sum_{i=1}^{K} \mathbf{1}\{y_i \neq y_{\sigma(i)}'\} = |S - S'|.$$

The average reward is therefore $\frac{1}{K}\sum_i r_i^{\text{OT}} = |S - S'|/K$. To compute its expectation, we analyze $|S - S'|$. Write the difference as a sum of centered terms:

$$S - S' = \sum_{i=1}^{K}\left(y_i - \tfrac{1}{2}\right) - \sum_{j=1}^{K}\left(y_j' - \tfrac{1}{2}\right),$$

where each summand $(y_i - \frac{1}{2})$ or $(y_j' - \frac{1}{2})$ equals $\pm\frac{1}{2}$ with equal probability. This is a sum of $2K$ independent, mean-zero, bounded random variables. The variance is:

$$\text{Var}(S - S') = K \cdot \frac{1}{4} + K \cdot \frac{1}{4} = \frac{K}{2}.$$

By the central limit theorem:

$$\frac{S - S'}{\sqrt{K/2}} \xrightarrow{d} \mathcal{N}(0, 1).$$

For a random variable $Z \sim \mathcal{N}(0, \sigma^2)$, we have $\mathbb{E}|Z| = \sigma\sqrt{2/\pi}$. Taking $\sigma = \sqrt{K/2}$ gives:

$$\mathbb{E}|S - S'| \approx \sqrt{\frac{K}{2}} \cdot \sqrt{\frac{2}{\pi}} = \sqrt{\frac{K}{\pi}}.$$

Therefore:

$$\mathbb{E}[r_i^{\text{OT}}] = \frac{\mathbb{E}|S - S'|}{K} \approx \frac{1}{\sqrt{\pi K}}.$$

*Table 2.* Expected per-completion reward under the random baseline (uninformative model).

| Strategy | $\mathbb{E}[r_i]$ | Scaling with $K$ |
| --- | --- | --- |
| Random Pairing | $\mathbb{E}[r_i^{\text{rand}}] = 1/2$ | Constant |
| One-to-All | $\mathbb{E}[r_i^{\text{all}}] = 1/2$ | Constant |
| Minimal Consistency | $\mathbb{E}[r_i^{\text{OT}}] \approx 1/\sqrt{\pi K}$ | $O(1/\sqrt{K})$ |

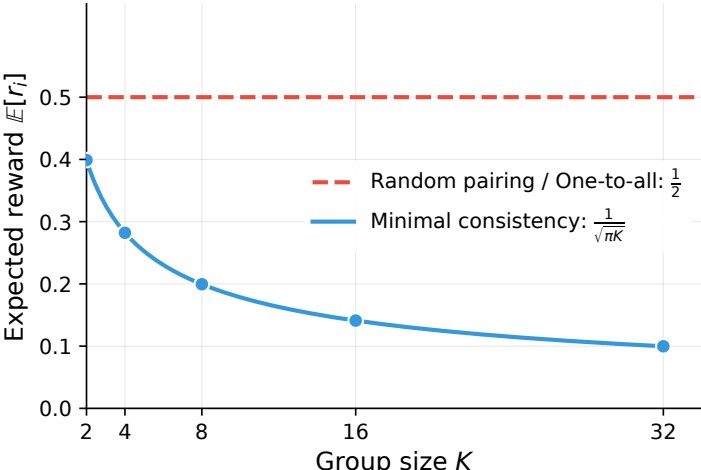

*Figure 7.* Expected per-completion reward under the random baseline as a function of group size $K$. Random pairing and one-to-all remain at $1/2$ regardless of $K$, while minimal consistency decays as $1/\sqrt{\pi K}$.

### A.4. Summary and Discussion

Table 2 summarizes the expected per-completion reward under the random baseline for each pairing strategy.

Figure 7 visualizes this comparison. For typical group sizes ($K = 8$ to $32$), minimal consistency yields expected rewards between 0.20 and 0.10—well below the 0.50 baseline of the other strategies.

**Interpretation.** Only minimal consistency penalizes uninformative models: as $K$ grows, the expected reward for a random guesser vanishes. The intuition is that minimal consistency actively searches for the *worst-case* pairing. A random model produces roughly equal numbers of True and False completions in each group, so the optimal matching can pair most completions with identical answers (which fail the verifier). Only the "leftover" completions—those exceeding the count in the other group—contribute to the reward, and this imbalance scales as $\sqrt{K}$ rather than $K$.

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

box.                                          box.
- object 2 = "car", highlighted by a blue     - object 2 = "car", highlighted by a blue
box.                                          box.
Is object 1 closer to the camera than         Is object 1 further from the camera than
object 2?                                      object 2?
```
**Answer: True**                              **Answer: False**

*Figure 12.* KITTI depth task example. The augmented prompt applies color jitter and relation swap ("closer to" → "further from"), but no horizontal flip. Since relation swap is a single equivariant transformation (one negation), the **answers should differ**.

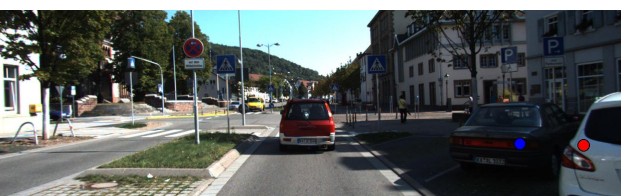 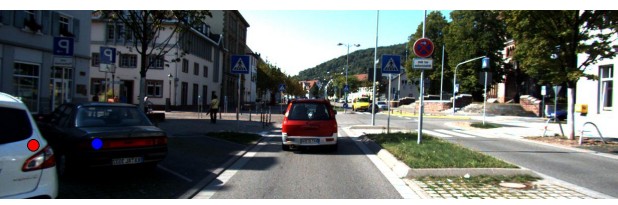

```
- object 1 = "car", marked with a red dot.    - object 1 = "car", marked with a red dot.
- object 2 = "car", marked with a blue dot.   - object 2 = "car", marked with a blue dot.
Is object 1 to the left of object 2?          Is object 1 to the right of object 2?
```
**Answer: False**                             **Answer: False**

*Figure 13.* KITTI orientation task example. The augmented prompt applies horizontal flip, color jitter, and relation swap ("left of" → "right of"). The flip negates the spatial relationship, and the relation swap negates the question—two equivariant transformations that cancel out, so the **answers should match**.

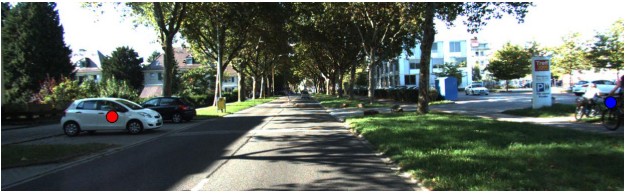 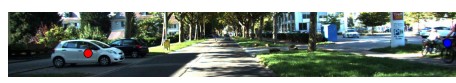

```
- object 1 = "car", marked with a red dot.    - object 1 = "car", marked with a red dot.
- object 2 = "cyclist", marked with a blue    - object 2 = "cyclist", marked with a blue
dot.                                          dot.
Is object 1 bigger than object 2?             Compared to object 2, is object 1 larger?
```
**Answer: False**                             **Answer: False**

*Figure 14.* KITTI size task example. The augmented prompt applies a bounding-box-aware crop, color jitter, and a different question template—but no relation swap. All transformations are invariant (zero negations), so the **answers should match**.

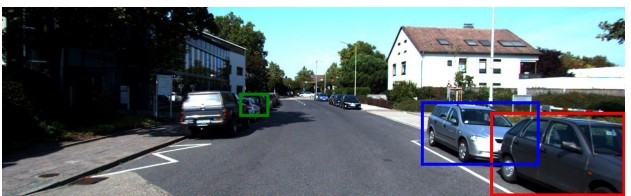

- object 1 = "car", highlighted by a red box.
- object 2 = "car", highlighted by a blue box.
- object 3 = "car", highlighted by a green box.

Is object 2 closer to object 1 than object 3?

**Answer: True**

- object 1 = "car", highlighted by a red box.
- object 2 = "car", highlighted by a blue box.
- object 3 = "car", highlighted by a green box.

Is object 2 further from object 1 than object 3?

**Answer: False**

*Figure 15.* KITTI relative distance task example (triplet). The augmented prompt applies horizontal flip, color jitter, and relation swap ("closer to" → "further from"). The flip does not affect inter-object distances, so only the relation swap contributes a negation—the **answers should differ**.