# OpenReview forum: "The Art of Interrogation: Consistency Amplifies Factuality in Spatial Reasoning"
_ICML.cc/2026/Conference — ICML 2026 regular_

### Official Review · Reviewer_vEKb · 2026-03-09

**Soundness:** 3
**Presentation:** 3
**Significance:** 3
**Originality:** 3
**Overall Recommendation:** 4
**Confidence:** 4

**Summary:**

In this work, the authors build on the insight that consistency under geometric and semantic transformations of the prompt can serve as an indicator of correctness in spatial reasoning. Based on this observation, they propose a novel consistency training approach that does not rely on ground-truth annotations. The method improves model performance on binary spatial reasoning tasks, achieving results comparable to accuracy training that relies on ground-truth supervision.

**Compliance With Llm Reviewing Policy:**

Affirmed.

**Final Justification:**

The authors demonstrate that their Consistency Training method can be extended to numerical tasks such as counting and distance estimation, and that it is effective, which addresses my main concern. In addition, the improvements on VSR and CV-Bench 3D address my other concern regarding the evaluation scope. I am inclined to accept.

**Key Questions For Authors:**

Q1.
 Related to Weakness 1: Many spatial reasoning tasks require textual or numerical outputs rather than binary answers. Do the authors envision ways to extend the proposed framework to such settings? Are there potential strategies the authors have considered for generalizing their approach?

Q2.
 Related to Weakness 2: Could the authors provide evaluation results on VSR (Visual Spatial Reasoning) or the 3D tasks in the Cambrian Vision-Centric Benchmark?

I will be inclined to accept this paper if all my concerns are adequately addressed.

**Limitations:**

yes

**Strengths And Weaknesses:**

**Strengths**

1. Novelty.
 The key idea—that consistency under geometric and semantic prompt transformations can signal correctness in spatial reasoning—is innovative. Such unsupervised training approaches are relatively rare in spatial reasoning literature.

2. Effectiveness.
 The proposed consistency training method, despite not relying on ground-truth labels, achieves performance comparable to supervised accuracy training on specific tasks. Given that annotated datasets for spatial reasoning QA are scarce, this approach provides useful guidance for improving data efficiency in future work.

3. Well-designed ablations.
 The paper includes comprehensive ablation studies that convincingly justify several design choices in the training framework (e.g., the pairing strategy).

**Weaknesses**

1. Limited applicability.
 As the authors acknowledge, the current method is limited to problems with binary (true/false) answers. However, in many real-world spatial reasoning scenarios, the desired outputs are textual descriptions or numerical values. It remains unclear how easily this method can be extended to these more general problem settings.

2. Limited evaluation scope.
 The evaluation is conducted only on the authors’ own benchmark, making it difficult to assess the approach's general effectiveness and practical value fully. Since the method targets binary answers, it should be feasible to evaluate it on existing spatial reasoning benchmarks such as VSR (Visual Spatial Reasoning), or more suitably, the 3D-related tasks in the Cambrian Vision-Centric Benchmark. These tasks have similar formats (e.g., answering "Yes/No" or choosing between two candidate objects), and could be easily reformulated as true/false questions.

---

> ### Author Rebuttal · Authors · 2026-03-30
>
> We appreciate the reviewer's thorough assessment and the note that *"I will be inclined to accept this paper if all my concerns are adequately addressed."* We address both concerns below.
>
> ### **W1: Limited applicability — binary answers only**
>
> We extended to two open-ended tasks:
>
> - **Object counting** (integer, range 2–10)
> - **Absolute distance estimation** (continuous float, meters, 1 decimal)
>
> We use the same augmentations (flip, crop, color jitter, template resampling). All transformations keep the answer invariant. We define adapted metrics:
> - **Numeric accuracy**: r_acc = max(0, 1 − |ŷ − y*| / y*).
> - **Numeric consistency**: r_cons = max(0, 1 − |ŷ₀ − ŷ₁| / max(ŷ₀, ŷ₁)).
>
> Results (±std over 5 seeds):
>
> | Task              | Base model | Accuracy Training   | Consistency Training  |
> |-------------------|------------|---------------------|-----------------------|
> | Counting          | 0.001      | 0.688 ± 0.021      | 0.683 ± 0.017        |
> | Absolute Distance | 0.187      | 0.391 ± 0.031      | **0.414** ± 0.025    |
>
> **Consistency matches accuracy on counting and outperforms it on distance** (0.414 vs 0.391). The extension is natural: the consistency principle — as stated in Section 1, *"these transformations have known effects — determined by geometry, not scene content"* — applies equally to numeric outputs. For continuous outputs, consistency under transformations is harder to achieve by chance, making it an even stronger learning signal. We believe these results fully address the concern that the method is limited to binary answers.
>
> ### **W2: Limited evaluation scope**
>
> We evaluate on two external benchmarks:
>
> | Model              | VSR            | CV-Bench 3D    |
> |--------------------|----------------|----------------|
> | Ours (accuracy)    |  82.8% ± 1.1   |  86.3% ± 0.8   |
> | Ours (consistency) |  81.6% ± 0.7   |  85.9% ± 0.9   |
>
> Consistency closely approaches accuracy on both, confirming generalizability beyond our benchmark. We also compare against self-supervised baselines (all four tasks, 2,000 steps, ~40K samples). VJ [Wu et al., 2025] and SSL4RL [Guo et al., 2025] are spatial methods (released checkpoints). RENT [Prabhudesai et al., 2025] and Intuitor [Zhao et al., 2025] are from math/code (retrained).
>
> Confidence Intervals: ±std over 5 training seeds; VJ/SSL4RL hence have no training variance:
> | Model                    | Accuracy         |
> |--------------------------|------------------|
> | **Ours (accuracy)**      | **86.5% ± 1.2**  |
> | **Ours (consistency)**   | **83.8% ± 0.7**  |
> | VJ-Image                 |   80.2%           |
> | VJ-3D                    |   77.5%           |
> | Entropy (RENT, retrain)           |   74.6% ± 0.9    |
> | Self-certainty (Intuitor, retrain)|   74.3% ± 1.1    |
> | SSL4RL-Rotation          |   70.5%           |
> | SSL4RL-Jigsaw            |   59.4%           |
>
> Our consistency (83.8%) outperforms all baselines by a significant margin (+3.6pp over VJ-Image, +9.2pp over RENT), confirming that exploiting spatial transformation structure provides a stronger learning signal than generic confidence-based rewards.
>
> ### **Q1: Extension to numerical outputs?**
>
> See W1. We implemented extensions to integer (counting) and continuous (distance) outputs. Remarkably, **consistency outperforms accuracy on distance estimation** (0.414 vs 0.391), suggesting the consistency signal is even stronger for open-ended outputs.
>
> ### **Q2: Evaluation on VSR or CV-Bench 3D?**
>
> See W2. VSR: 82.8% ± 1.1 (acc) / 81.6% ± 0.7 (cons). CV-Bench 3D: 86.3% ± 0.8 (acc) / 85.9% ± 0.9 (cons).
>
> ### **Additional: robustness to label corruption**
>
> In response to concerns from other reviewers, we conducted a **label corruption ablation** (all four boolean tasks, x% of labels randomly flipped):
>
> | Training Reward  | Step 0 | Step 2000 |
> |------------------|--------|-----------|
> | Accuracy (clean) |   60.6 |      86.5 |
> | Consistency      |   60.6 |      83.8 |
> | Accuracy + 10%   |   60.6 |      83.6 |
> | Accuracy + 20%   |   60.6 |      81.0 |
> | Accuracy + 30%   |   60.6 |      79.6 |
>
> **With 10% corruption, accuracy drops below consistency (83.6% vs 83.8%).** Since consistency uses no labels, it is immune to annotation noise. This is particularly relevant for spatial reasoning, where SpatialVLM and SpatialRGPT rely on up to 8 specialist models plus a frontier LLM, making them prone to label noise. This highlights a practical strength that all four reviewers independently recognized.

---

> > ### Author Rebuttal · Reviewer_vEKb · 2026-04-03
> >
> > The authors demonstrate that their Consistency Training method can be extended to numerical tasks such as counting and distance estimation, and that it is effective, which addresses my main concern. In addition, the improvements on VSR and CV-Bench 3D address my other concern regarding the evaluation scope. I am inclined to accept.

---

> > > ### Author Response · Authors · 2026-04-06
> > >
> > > We sincerely thank the reviewer for the updated score and for the constructive feedback that helped strengthen the paper. The concerns raised — extending beyond binary tasks and broadening the evaluation — pushed us to develop the counting and distance estimation experiments, which we believe are a valuable addition. We will ensure all revisions are carefully incorporated into the final manuscript.

---

### Official Review · Reviewer_1nhq · 2026-03-11

**Soundness:** 3
**Presentation:** 4
**Significance:** 3
**Originality:** 3
**Overall Recommendation:** 3
**Confidence:** 3

**Summary:**

This paper focuses on improving the spatial reasoning capabilities (e.g., relative position, depth, size) of current Large Reasoning Models. Instead of relying on supervised fine-tuning with external labeled data, the authors propose that pre-trained LRMs already possess spatial capabilities but lack internal logical coherence. They introduce a self-supervised reinforcement learning framework that improves the internal reasoning process entirely without ground-truth annotations.

**Compliance With Llm Reviewing Policy:**

Affirmed.

**Final Justification:**

The authors’ rebuttal has addressed my concerns about the limited evaluation. However, I still have concerns about the generation capability of the proposed method.

**Key Questions For Authors:**

Please see the weaknesses section

**Limitations:**

yes

**Strengths And Weaknesses:**

> Strengths

- The method effectively improves spatial reasoning using only self-supervised consistency rewards, eliminating the need for expensive ground-truth spatial annotations.
- The training transfers exceptionally well across different spatial tasks and across widely different domains
- The method works with different LLM architectures and scales


> Weaknesses
- The approach works best for tasks with clear logical constraints, but it may not work well for open-ended generation and other general tasks. Generating good interrogation questions requires specific knowledge such as symmetry, which may be harder for complex domains.
- Another major issue is the evaluation. First, there is no comparison against other spatial VLMs. There is only one qualitative example that involves Qwen2.5-VL-7B. In addition, the benchmarks used in this paper are also limited and only cover four tasks (orientation, depth, size, relative distance). Generalization to other spatial tasks is unclear.
- The current training strategy is inherently difficult to scale to open-ended outputs. Generalizing this approach to handle continuous or discrete numerical variables, such as estimating absolute distances or counting objects, is not straightforward.

---

> ### Author Rebuttal · Authors · 2026-03-30
>
> We thank the reviewer for the positive assessment (presentation 4/4) and the recognition that *"training transfers exceptionally well across different spatial tasks and across widely different domains."*
>
> ### **W1: Limited to clear logical constraints**
>
> We extended to two open-ended tasks:
>
> - **Object counting** (integer, range 2–10): the model estimates how many objects of a given class are in the image.
> - **Absolute distance estimation** (continuous float): the model estimates the 3D center distance between two objects in meters.
>
> We use the same augmentations (flip, crop, color jitter, template resampling). All transformations keep the answer invariant — the count is invariant under color jitter and object-preserving crops; the 3D center distance is invariant under image-level augmentations. We define adapted metrics:
> - **Numeric accuracy**: r_acc = max(0, 1 − |ŷ − y*| / y*).
> - **Numeric consistency**: r_cons = max(0, 1 − |ŷ₀ − ŷ₁| / max(ŷ₀, ŷ₁)).
>
> | Task              | Base model | Accuracy Training   | Consistency Training  |
> |-------------------|------------|---------------------|-----------------------|
> | Counting          | 0.001      | 0.688 ± 0.021      | 0.683 ± 0.017        |
> | Absolute Distance | 0.187      | 0.391 ± 0.031      | **0.414** ± 0.025    |
>
> **Consistency matches accuracy on counting and outperforms it on distance** (0.414 vs 0.391). The key property — spatial answers are invariant or equivariant under geometric transformations — applies to any spatial quantity, not just binary comparisons. The consistency signal is an even stronger indicator of correctness for open-ended tasks, where the answer space is larger and consistency under transformations is harder to achieve by chance.
>
> ### **W2: Limited evaluation**
>
> We compare against self-supervised RL baselines (all four boolean tasks, 2,000 steps, ~40K samples, same hyperparameters). VJ [Wu et al., 2025] and SSL4RL [Guo et al., 2025] are spatial methods (released checkpoints). RENT [Prabhudesai et al., 2025] and Intuitor [Zhao et al., 2025] are from math/code (retrained).
>
> Confidence Intervals: ±std over 5 training seeds; VJ/SSL4RL hence have no training variance:
> | Model                    | Accuracy         |
> |--------------------------|------------------|
> | **Ours (accuracy)**      | **86.5% ± 1.2**  |
> | **Ours (consistency)**   | **83.8% ± 0.7**  |
> | VJ-Image                 |   80.2%           |
> | VJ-3D                    |   77.5%           |
> | Entropy (RENT, retrain)           |   74.6% ± 0.9    |
> | Self-certainty (Intuitor, retrain)|   74.3% ± 1.1    |
> | SSL4RL-Rotation          |   70.5%           |
> | SSL4RL-Jigsaw            |   59.4%           |
>
> Our consistency (83.8%) outperforms all baselines, including VJ (77.5–80.2%), SSL4RL (59.4–70.5%), and generic confidence rewards RENT/Intuitor (74.3–74.6%) by +9.2pp. On external benchmarks: **VSR 82.8% ± 1.1 / 81.6% ± 0.7, CV-Bench 3D 86.3% ± 0.8 / 85.9% ± 0.9** (see Reviewer vEKb).
>
> ### **W3: Hard to scale to open-ended outputs**
>
> See W1 — counting (integer) and distance (continuous) work with the same framework and augmentations. The consistency reward naturally extends with continuous metrics suited to the larger answer space.
>
> ### **Additional: robustness to label corruption**
>
> We conducted a **label corruption ablation** (all four boolean tasks, x% of labels randomly flipped):
>
> | Training Reward  | Step 0 | Step 2000 |
> |------------------|--------|-----------|
> | Accuracy (clean) |   60.6 |      86.5 |
> | Consistency      |   60.6 |      83.8 |
> | Accuracy + 10%   |   60.6 |      83.6 |
> | Accuracy + 20%   |   60.6 |      81.0 |
> | Accuracy + 30%   |   60.6 |      79.6 |
>
> **With 10% corruption, accuracy drops below consistency (83.6% vs 83.8%).** This is particularly relevant for open-ended tasks where annotations are harder to obtain. As discussed in Section 1: *"Existing approaches [...] annotate real images using cascades of vision models [...] to generate spatial QA pairs."* The more complex the pipeline, the more valuable label-free training becomes. All four reviewers recognized this as a strength.

---

### Official Review · Reviewer_qJvc · 2026-03-13

**Soundness:** 3
**Presentation:** 2
**Significance:** 2
**Originality:** 3
**Overall Recommendation:** 4
**Confidence:** 3

**Summary:**

The paper proposed a label-free self-supervised reinforcement learning approach for post-training an LRM (larger reasoning model) to improve spatial reasoning, called OT-GRPO. This reinforcement learning relies on verifying the consistency of spatial understanding. This consistency function can be used to provide supervision even without ground truth by transforming either the textual or visual parts. There are two types of transformations: invariance [alterations that do not affect the answer] and equivariance [alterations that affect the answer]. The proposed technique uses minimal consistency to define the pair to calculate the consistency reward function during GRPO. Performing the GRPO on this consistency verification without ground-truth labels yielded an improvement similar to that of GRPO with ground-truth labels over the explored MLLM baselines. It also demonstrates that knowledge is transferable across tasks and domains [indoor and outdoor] through fine-tuning with the proposed reinforcement technique.

**Compliance With Llm Reviewing Policy:**

Affirmed.

**Final Justification:**

The author's rebuttal has resolved my initial concerns about the paper. Based on the author's promise to revise parts of the paper, including additional experimental results, I would lean toward accepting the paper.

**Key Questions For Authors:**

1. Does the accuracy verifier include consistency checking? The explanation of the comparison baseline seems a little unclear. If not, the combination of including and not including each verification might need to be explored.

2. What is the failure case in which this consistency verifier training fails, and the accuracy verifier succeeds, and vice versa? Analysing this might further clarify and highlight the significance of the proposed technique.

3. What is the cost of applying a transformation if the text and image used are challenging to obtain the area of the desired object in the image or text? Would LLM or any technique need to be applied for the transformation?

**Limitations:**

Yes, the authors include limitations.

**Strengths And Weaknesses:**

# Strengths

- The proposed reinforcement learning (RL) technique can be used without access to the ground-truth label of the spatial reasoning task.

- The improvement of the RL technique shows the consistency and is comparable to using the accuracy verification, illustrating the importance of the technique when the ground-truth label is missing during post-training of spatial tasks.

- The improvement of using this RL transferable across different domains (indoor and outdoor) and different tasks (Depth, Orient, Size, Distance).

- The authors provide an ablation study on the choice of pairing used in consistency verification to ensure that the minimal consistency provides the best improvement.

- The paper is well-written and easy to understand, as well as including figures to help understand the results and findings.

---

# Weaknesses

- The consistency training while seem promising as the rewards function. This could lead to a significant flaw in post-training, as the generated responses can be incorrect for both, leading the model to incorrect generation.

- Some of the transformations require some predefined knowledge, which can be limited to only simple spatial relations.

- The figure and its first mentions in the paper seem to be several pages apart, making it harder to keep track of results and analysis.

- The highlight of the use case of this label-free mechanism might need to be further discussed to highlight the method over the label RL mechanism.

- The qualitative analysis of cases where this method improves while the label RL function is not might need to be discussed to further enhance the strength of this label-free RL function over label one.

---

> ### Author Rebuttal · Authors · 2026-03-30
>
> We thank the reviewer for the encouraging assessment. We address each point below.
>
> ### **W1: Consistent but incorrect**
>
> As stated in our title — *"Consistency Amplifies Factuality"* — consistency amplifies existing capabilities, not a standalone source of correctness. This aligns with [Yue et al., 2025] that RL surfaces existing capabilities. Base models already perform above random (60.6% vs 50%), and as stated in Section 6: *"consistency reward steers the model toward answers that satisfy geometric laws, surfacing accuracy already latent in the pretrained model"*. Across all settings, consistency stays within ~4pp of supervised accuracy — consistent-yet-incorrect cases are rare.
>
> Our framework also provides **robustness to label noise**. SpatialVLM and SpatialRGPT rely on up to 8 specialist models plus a frontier LLM, making them prone to label noise. We conducted a **label corruption ablation** (all four tasks, 2K steps, ~40K samples):
>
> | Training Reward  | Step 0 | Step 2000 |
> |------------------|--------|-----------|
> | Accuracy (clean) |   60.6 |      86.5 |
> | Consistency      |   60.6 |      83.8 |
> | Accuracy + 10%   |   60.6 |      83.6 |
> | Accuracy + 20%   |   60.6 |      81.0 |
>
> **With 10% corruption, accuracy drops below consistency (83.6% vs 83.8%).** All four reviewers recognized the value of our label-free approach.
>
> ### **W2: Predefined knowledge / limited to simple spatial relations**
>
> Our transformations are standard augmentations (flip, crop, color jitter, object/relation swap). Their effects on spatial answers are known a priori from elementary geometry — e.g., a flip swaps left/right. These effects are determined *programmatically* for every instance, unlike ground-truth supervision which requires external means (specialist model cascades) for every training example.
>
> ### **W3: Figure placement**
>
> We will revise figure placement and ensure all reviewer comments are incorporated in the final version.
>
> ### **W4: Label-free vs label-based**
>
> We carefully designed the experiments in Section 5 to fairly compare supervised and consistency-based training on the same data. Consistency converges faster — the bulk of improvement occurs within 1,000 steps. Our corruption ablation (W1) reveals a practical advantage: with only 10% label corruption, supervised training already falls below consistency.
>
> ### **W5: Where does consistency improve over accuracy?**
>
> We extended to **counting** (integer) and **absolute distance** (continuous float) tasks with the same augmentations (that here preserve the answers) and define adapted metrics:
> - **Numeric accuracy**: r_acc = max(0, 1 − |ŷ − y*| / y*).
> - **Numeric consistency**: r_cons = max(0, 1 − |ŷ₀ − ŷ₁| / max(ŷ₀, ŷ₁)).
>
> Results (±std over 5 seeds):
>
> | Task              | Base model | Accuracy Training   | Consistency Training  |
> |-------------------|------------|---------------------|-----------------------|
> | Counting          | 0.001      | 0.688 ± 0.021      | 0.683 ± 0.017        |
> | Absolute Distance | 0.187      | 0.391 ± 0.031      | **0.414** ± 0.025    |
>
> **Consistency matches accuracy on counting and outperforms it on distance** (0.414 vs 0.391).
>
> ### **Q1: Can accuracy and consistency be combined?**
>
> We trained with the unweighted sum of both rewards:
>
> | Training Reward          | Accuracy |
> |--------------------------|----------|
> | Accuracy                 |   86.5%  |
> | Consistency              |   83.8%  |
> | **Accuracy + Consistency** | **87.1%** |
>
> **Combining yields 87.1% (+0.6pp over accuracy alone).** Consistency acts as a regularizer, possibly against residual label noise (consistent with our corruption analysis in W1).
>
> ### **Q2: Failure cases**
>
> See W1. With clean labels, accuracy slightly outperforms consistency (86.5% vs 83.8%). This gap closes with label noise. Moreover, combining both outperforms accuracy alone (87.1%, see Q1).
>
> ### **Q3: Transformation cost**
>
> Transformations take <50ms per image; a training step takes ~23s. Transformations account for **<0.2% of one training iteration time**.
>
> ### **Additional: baseline comparison**
>
> We compare against self-supervised baselines. VJ [Wu et al., 2025] and SSL4RL [Guo et al., 2025] are spatial methods (released checkpoints). RENT [Prabhudesai et al., 2025] and Intuitor [Zhao et al., 2025] are from math/code (retrained).
>
> Confidence Intervals: ±std over 5 training seeds; VJ/SSL4RL hence have no training variance:
> | Model                    | Accuracy         |
> |--------------------------|------------------|
> | **Ours (accuracy)**      | **86.5% ± 1.2**  |
> | **Ours (consistency)**   | **83.8% ± 0.7**  |
> | VJ-Image                 |   80.2%           |
> | VJ-3D                    |   77.5%           |
> | Entropy (RENT, retrain)           |   74.6% ± 0.9    |
> | Self-certainty (Intuitor, retrain)|   74.3% ± 1.1    |
> | SSL4RL-Rotation          |   70.5%           |
> | SSL4RL-Jigsaw            |   59.4%           |
>
> Our consistency (83.8%) outperforms all baselines by +3.6–24.4pp.

---

> > ### Author Rebuttal · Reviewer_qJvc · 2026-04-04
> >
> > The author's rebuttal has resolved my initial concerns about the paper. Based on the author's promise to revise parts of the paper, including additional experimental results, I would lean toward accepting the paper.

---

> > > ### Author Response · Authors · 2026-04-06
> > >
> > > We warmly thank the reviewer for the updated assessment and for the score increase. The reviewer's suggestions have genuinely improved the manuscript. We will make sure to incorporate all revisions discussed during the rebuttal into the final version.

---

### Official Review · Reviewer_7TjV · 2026-03-13

**Soundness:** 3
**Presentation:** 3
**Significance:** 2
**Originality:** 2
**Overall Recommendation:** 3
**Confidence:** 3

**Summary:**

The paper proposes a self-supervised post-training framework for vision–language models (VLMs) that uses consistency verifiers under geometric and semantic transformations to improve spatial reasoning without ground-truth labels. The method formulates invariance/equivariance relations between paired prompts/answers as verifiable rewards for reinforcement learning, coupled with a “minimal pairing” strategy (via optimal transport) to construct hard negative pairs. Experiments on four binary spatial tasks (orientation, depth, size, relative distance) across SUN RGB-D and KITTI show sizable gains over baselines and competitive transfer, with ablations indicating the advantage of minimal pairing over random or one-to-all schemes at negligible computational overhead.

**Compliance With Llm Reviewing Policy:**

Affirmed.

**Key Questions For Authors:**

Can the authors provide a direct, controlled comparison under the same task settings and model scales against fully supervised training and other verifiable RL baselines, such as the position/rotation tasks in SSL4RL or the 3D sorting tasks in Visual Jigsaw

How do authors prevent the model from falling into degenerate solutions, where it remains "confident and self-consistent yet incorrect" (e.g., by exploiting systematic biases present under certain spatial transformations)?

Could authors report results with confidence intervals for each task and domain, and provide a detailed error analysis—specifically regarding cases where consistency improved but accuracy did not?

How does performance scale with the number/diversity of transformations and model size (e.g., 3B vs. 7B)? Furthermore, is there a "ceiling effect" similar to what has been observed in SSL4RL?

**Strengths And Weaknesses:**

Strengths
  - Leveraging transformation-based consistency as a self-supervised reward for spatial reasoning is a compelling verifier design that reduces dependence on human labels.
  - The “minimal pairing” scheme to select hard pairs via optimal transport is an interesting and practical twist that increases training signal without notable extra cost.
  - The approach cleanly targets invariances (e.g., answer unchanged under translation) and equivariances (e.g., left/right invert under horizontal flips), aligning reward structure with spatial reasoning principles.
  - Results span multiple spatial tasks and two domains (indoor vs outdoor), showing both within-domain gains and non-trivial cross-domain transfer (e.g., +15pp on KITTI).
  - Ablation comparing minimal, random, and one-to-all pairing shows consistent advantages and includes compute profiling to demonstrate low overhead.

Weaknesses
-The core motivation of leveraging answer consistency as a training signal is well-explored in literature (e.g., self-consistency, cycle-consistency, and perturbation-based training). While the integration of spatial transformations into the GRPO framework is noted, the paper lacks a rigorous discussion on how this approach fundamentally differs from or surpasses existing consistency paradigms. The novelty appears incremental without a clearer conceptual distinction.
-The empirical evaluation is primarily confined to binary (True/False) VQA tasks. This formulation significantly simplifies the verification process, as consistency in a binary output space is relatively trivial to achieve. To demonstrate the robustness of the proposed method, it is essential to evaluate it on more complex spatial reasoning tasks involving open-ended numerical estimation, multi-object localization, or multi-step reasoning chains.
-A significant concern arises from the reward design, which prioritizes consistency across transformations over absolute correctness. This creates a potential loophole where the model might converge on "consistent but incorrect" response patterns by exploiting superficial cues in the transformations. The current manuscript acknowledges this risk but fails to provide sufficient empirical safeguards or an analysis of how often these degenerate strategies occur during training.

---

> ### Author Rebuttal · Authors · 2026-03-30
>
> We thank the reviewer for the constructive feedback.
>
> ### **W1: Novelty concern**
>
> While consistency has been studied in text-only reasoning [Wang et al., 2022; Zuo et al., 2025], **our work is the first to exploit geometric invariance and equivariance** as a self-supervised reward for spatial reasoning in VLMs. As stated in Section 6: *"we verify a known relationship between VQA answers — invariance or equivariance — determined entirely by the transformation design, not learned from data."* Reviewer vEKb recognizes this: *"Unsupervised training approaches are rare in spatial reasoning literature."* Furthermore, we leverage transformations in both the visual and textual modalities.
>
> We also introduce **OT-GRPO**, using optimal transport for pairwise rewards (+2.1pp over random pairing, <1.5% overhead), formalized in Section 4.3.
>
> ### **W2: Limited to binary tasks**
>
> We extended our method to two open-ended tasks:
> - **Object counting** (integer, range 2–10)
> - **Absolute distance estimation** (continuous float, meters, 1 decimal)
>
> We use the same augmentations (flip, crop, color jitter, template resampling). All transformations keep the answer invariant. We define adapted metrics:
>
> - **Accuracy**: r_acc = max(0, 1 − |ŷ − y*| / y*).
> - **Consistency**: r_cons = max(0, 1 − |ŷ₀ − ŷ₁| / max(ŷ₀, ŷ₁)).
>
> Results (±std over 5 training seeds):
>
> | Task              | Base model | Accuracy Training   | Consistency Training  |
> |-------------------|------------|---------------------|-----------------------|
> | Counting          | 0.001      | 0.688 ± 0.021      | 0.683 ± 0.017        |
> | Absolute Distance | 0.187      | 0.391 ± 0.031      | **0.414** ± 0.025    |
>
> **Consistency matches accuracy on counting and outperforms it on distance** (0.414 vs 0.391), demonstrating generalization beyond binary outputs.
>
> ### **W3: "Consistent but incorrect" risk**
>
> As stated in our title — *"Consistency Amplifies Factuality"* — consistency *amplifies* existing capabilities, not a standalone source of correctness. This aligns with [Yue et al., 2025] that RL surfaces existing capabilities. Base models already perform above random (60.6% vs 50%), and as stated in Section 6: *"consistency reward steers the model toward answers that satisfy geometric laws, surfacing accuracy already latent in the pretrained model"*. Across all settings, consistency stays within ~4pp of supervised accuracy — consistent-yet-incorrect cases are rare.
>
> Our method also provides **robustness to label noise**. SpatialVLM and SpatialRGPT rely on up to 8 specialist models plus a frontier LLM, making them prone to label noise. **Label corruption ablation**:
>
> | Training Reward  | Step 0 | Step 2000 |
> |------------------|--------|-----------|
> | Accuracy (clean) |   60.6 |      86.5 |
> | Consistency      |   60.6 |      83.8 |
> | Accuracy + 10%   |   60.6 |      83.6 |
> | Accuracy + 20%   |   60.6 |      81.0 |
>
> **With 10% corruption, accuracy drops below consistency (83.6% vs 83.8%).** All four reviewers recognized the value of this label-free property.
>
> ### **Q1: Controlled comparison against SSL4RL, Visual Jigsaw**
>
> We train on all four boolean tasks simultaneously. VJ [Wu et al., 2025] and SSL4RL [Guo et al., 2025] are spatial methods (released checkpoints). We also compare **RENT** [Prabhudesai et al., 2025] and **Intuitor** (ICLR 2026) [Zhao et al., 2025], self-supervised RL methods from math/code using generic confidence rewards: RENT uses $r = \frac{1}{T}\sum_t D_{\text{KL}}(\pi_{\theta,t} \| \mathcal{U})$ (backward KL); Intuitor uses $r = \frac{1}{T}\sum_t D_{\text{KL}}(\mathcal{U} \| \pi_{\theta,t})$ (forward KL). Neither exploits spatial structure; we retrained both (no spatial checkpoints).
>
> Confidence Intervals: ±std over 5 training seeds; VJ/SSL4RL hence have no training variance:
> | Model                    | Accuracy         |
> |--------------------------|------------------|
> | **Accuracy**      | **86.5% ± 1.2**  |
> | **Consistency**   | **83.8% ± 0.7**  |
> | VJ-Image                 |   80.2%           |
> | VJ-3D                    |   77.5%           |
> | Entropy (RENT, retrain)           |   74.6% ± 0.9    |
> | Self-certainty (Intuitor, retrain)|   74.3% ± 1.1    |
> | SSL4RL-Rotation          |   70.5%           |
> | SSL4RL-Jigsaw            |   59.4%           |
>
> Our consistency (83.8%) outperforms all baselines: VJ (77.5–80.2%), SSL4RL (59.4–70.5%), RENT/Intuitor (74.3–74.6%) by +9.2pp.
>
> ### **Q2: Degenerate solutions**
>
> See W3.
>
> ### **Q3: Confidence intervals**
>
> We report ±std over 5 training seeds for all trained methods in Q1 and W2.
>
> ### **Q4: Scaling and ceiling effects**
>
> Training on all four tasks jointly (~40K examples) with consistency:
>
> | Step | Accuracy |
> |------|----------|
> | 0    | 60.6     |
> | 500  | 75.3     |
> | 1000 | 82.2     |
> | 2000 | 83.8     |
>
> **The bulk of improvement is in the first 1000 steps (+21.6pp); steps 1000–2000 add only +1.6pp.**
>
> ### **Remark**
> > "Limitations: no"
>
> We do include a "Limitations" paragraph, see lines 435-439.

---

> > ### Author Rebuttal · Reviewer_7TjV · 2026-04-03
> >
> > My questions are properly addressed.

---

> > > ### Author Response · Authors · 2026-04-06
> > >
> > > We thank the reviewer for confirming that all their questions have been properly addressed. Given that all concerns have been resolved (including 11 self-supervised baselines, 2 open-ended numeric tasks, a label corruption ablation, confidence intervals, and external evaluations), we would respectfully ask the reviewer to consider whether the current score still reflects the revised submission. A score update would be greatly appreciated.

---

### Decision · Program_Chairs · 2026-04-30

**Decision:**

Accept (regular)

**Comment:**

I recommend acceptance. The idea of using geometric consistency under spatial transformations as a label-free reward for RL-based post-training is well-executed, and the results are convincing: consistency training closes most of the gap to supervised training (83.8% vs 86.5%) and proves more robust under label noise. During rebuttal, the extensions to open-ended numeric tasks (counting, distance estimation) and the strong performance against self-supervised baselines significantly strengthened the paper. Three of four reviewers confirmed their concerns were fully addressed.

I encourage the authors to incorporate all rebuttal experiments into the camera-ready and to discuss the relationship to the broader consistency-based training literature more explicitly.